# Characterization Methods along the Process Chain of Electrical Steel Sheet—From Best Practices to Advanced Characterization

**DOI:** 10.3390/ma15010032

**Published:** 2021-12-21

**Authors:** Martin Heller, Anett Stöcker, Rudolf Kawalla, Nora Leuning, Kay Hameyer, Xuefei Wei, Gerhard Hirt, Lucas Böhm, Wolfram Volk, Sandra Korte-Kerzel

**Affiliations:** 1Institute of Physical Metallurgy and Materials Physics (IMM), RWTH Aachen University, 52074 Aachen, Germany; korte-kerzel@imm.rwth-aachen.de; 2Institute of Metal Forming (IMF), TU Bergakademie Freiberg, 09596 Freiberg, Germany; anett.stoecker@imf.tu-freiberg.de (A.S.); rudolf.kawalla@imf.tu-freiberg.de (R.K.); 3Institute of Electrical Machines (IEM), RWTH Aachen University, 52052 Aachen, Germany; nora.leuning@iem.rwth-aachen.de (N.L.); kay.hameyer@iem.rwth-aachen.de (K.H.); 4Institute of Metal Forming (IBF), RWTH Aachen University, 52056 Aachen, Germany; xuefei.wei@ibf.rwth-aachen.de (X.W.); gerhard.hirt@ibf.rwth-aachen.de (G.H.); 5Chair of Metal Forming and Casting (utg), TU München, 85748 Garching, Germany; lucas.boehm@utg.de (L.B.); wolfram.volk@utg.de (W.V.)

**Keywords:** characterization, nano-/micromechanics, microstructure, recrystallization, XRD, magnetic properties, electrical steel sheet

## Abstract

Non-oriented (NO) electrical steel sheets find their application in rotating electrical machines, ranging from generators for wind turbines to motors for the transportation sector and small motors for kitchen appliances. With the current trend of moving away from fossil fuel-based energy conversion towards an electricity-based one, these machines become more and more important and, as a consequence, the leverage effect in saving energy by improving efficiency is huge. It is already well established that different applications of an electrical machine have individual requirements for the properties of the NO electrical steel sheets, which in turn result from the microstructures and textures thereof. However, designing and producing tailor-made NO electrical steel sheet is still challenging, because the complex interdependence between processing steps, the different phenomena taking place and the resulting material properties are still not sufficiently understood. This work shows how established, as well as advanced and newly developed characterization methods, can be used to unfold these intricate connections. In this context, the respective characterization methods are explained and applied to NO electrical steel as well as to the typical processing steps. In addition, several experimental results are reviewed to show the strengths of the different methods, as well as their (dis)advantages, typical applications and obtainable data.

## 1. Introduction—Characterization as the Basis for Tailor-Made Materials

In the course of its more than 100-year history, NO electrical steel sheet has become a high performance and essential material in today’s society. This is because the efficiency of rotating electrical machines, in which thin, stacked electrical steel sheets are mainly used to guide and magnify the magnetic flow, is key to reducing the overall environmental impact, which, on the other hand, is important in view of climate change and a rising demand for “clean” electrical machines (transportation—electric vehicles, energy sector—generators of wind turbines, household machines—mixers, etc.). Higher efficiencies can be achieved through tailor-made materials and microstructures matching the respective applications as well as through the general minimization of iron loss. Tailor-made materials and microstructures are especially important because rotation frequencies of the respective electrical machines have a large impact on the magnitude and share of the different iron loss components (hysteresis, classical and excess losses), which, in turn, depend on different material and microstructure properties like alloying elements, grain size, texture, sheet thickness or residual stresses. To make things even more complicated, many of these parameters have the opposite effect on the magnetizability itself [1,2]. The above-mentioned interrelationships are summarised in Table 1. Consequently, there is always the need for a trade-off between minimizing losses and maximizing magnetizability. For example, the generator of a wind turbine (low frequency) requires a different material and microstructure than the motor of an electric car (high frequency). Based on Table 1, in the latter case the focus should be on a small grain size and sheet thickness as well as on a high electrical resistivity, as the classical and excess losses dominate, while in the former case the hysteresis loss dominates, which can be reduced by a larger grain size (see [3] for more details).

In order to create such a tailor-made material or microstructure, the requirements of the electrical machine, the influence of different material properties on different loss components and how to adjust these properties accordingly along the process chain (see [4] for more details) need to be known. Behind all this knowledge are the characterization methods.

The work at hand is part of the special issue “Low-loss non-oriented electrical steel sheet for energy-efficient electrical drives”, in whose publications the same material is was used throughout (see chemical composition in Table 2). Thereby, every publication focuses on a different aspect of this material. Two papers revolve around processing and its influence on microstructure, texture and magnetic properties [4,5]; one deals with integrated process simulation [6] and the last with material design [3], all drawing heavily on experimental results from the established and advanced methods described and discussed here. In the current paper, we present the key methods for fully characterizing the most important properties of electrical steel. In this context, we discuss material-dependent considerations specific to electrical steel, as well as interpretation directions for different results and a classification according to the required effort. Besides simple methods like light microscopy (Section 3.1) and hardness measurements (Section 3.2), we also introduce advanced methods like electron backscatter diffraction (EBSD) (Section 4.8) and crystal growth (Section 4.2). Especially for the advanced methods, we aim to point out how each can help to increase the fundamental understanding of the material, and how this can be linked to possible efficiency improvements in the future. In addition, we present new selected experimental results to show the strengths of the respective methods using Fe2.4wt.%Si and Fe3.2wt.%Si as example materials (Table 2). Some of the results from characterization can directly be used to either run or improve physically based simulation models of the different process steps (see [6] for more details). Lastly, we summarise and reflect on the use of all methods in terms of an overview that provides a rating of all methods with respect to their significances for electrical steel and their connected processing steps, the effort needed to conduct the analyses and the insights gained from each method.

## 2. Typical Design Strategies and Challenges for NO Electrical Steel Sheet

Currently, a typical NO electrical steel sheet has a thickness of around 0.35 mm, a grain size of 20–200 µm, a silicon content of around 3 wt.% and an aluminium content of around 0.75 wt.% [7,8]. Since the texture can presently be influenced only to a limited extent on an industrial scale, it is mainly considered in research, although it is demonstrably an important influencing factor [9]. Both silicon and aluminium increase the electric resistivity of electrical steel, resulting in a reduction of the classical loss and in maximum polarization [10]. The latter point, in combination with a reduced formability when the silicon content is increased, leads to a maximum commercial silicon content of 3.5 wt.% [7]. At a silicon content of 1.9 wt.% and above, the phase transformation from body-centred cubic (bcc) to face-centred cubic (fcc), as observed in pure iron, will be suppressed along the process chain [11]. Hence, every process step has an influence on the final microstructure and texture and, therefore, on the resulting magnetic and mechanical properties. The main process and production steps after casting are hot rolling, cold rolling, final annealing and blanking. During final annealing the material’s microstructure is rebuilt from scratch through recovery and recrystallization. However, it has been shown that the number of present nuclei, as well as the kinetics of nucleus formation and growth, are heavily dependent on the rolling conditions and the resulting microstructures [12]. These characteristics, and thus each preceding process step, are therefore important, since, for the same heat treatment parameters, a high growth/formation rate ratio leads to a higher grain size (Table 1: Ph ↓, Pex ↑, J ↓, σy ↓) compared to a low growth/formation rate ratio. Moreover, although the nucleus formation process itself is not fully understood yet, it has been shown that formation and growth are not fully random, since specific rolling textures always result in specific recrystallization textures [13]. Finally, different studies [14,15] have shown that grain size is an important factor during blanking; although, here too, the underlying mechanisms have not yet been fully resolved. For example, intuitively and according to the Hall–Petch relationship one could assume that cutting becomes easier with increasing grain size; however, Kals et al. [16] summarised that the cutting work can increase again after reaching a specific grain-size-to-sheet-thickness ratio. The importance of this topic is further underlined in that, to reduce classical loss, the sheet thickness is then reduced to the same order of magnitude as the grain size (<0.3 mm). This means that just a few grains or even a single grain are cut along the sheet normal during the blanking process. Furthermore, dislocations are again introduced during this process step, which worsen the magnetic properties after a new microstructure with a very low defect density (dislocations and grain boundaries) was formed during recrystallization in the final annealing step.

A better understanding of the microstructure evolution and crystal plasticity of single to oligo- and polycrystals in electrical steel will therefore help to reveal the intricate connections between the individual processing steps, to achieve application-dependent targets for different microstructural parameters and to better connect the phenomena behind individual aspects of processing, such as changes in cutting work and properties in the final application, including the improvement and deterioration of the magnetic properties. The characterization methods presented here are the fundamental tools required to achieve these insights.

## 3. Best Practice for Standard Characterization Methods

Most methods in this section are standard methods, however, some specific factors need to be considered when dealing with electrical steel sheet to acquire the most important, material-specific information. Along with recommendations for best practice in using each method, we present new results and a range of possible interpretation directions for most methods.

### 3.1. Light Microscopy

Light microscopy is one of the most widespread methods in materials science. Nearly every lab or production facility dealing with material production, forming or analysis has a light microscope available. Generally, it gives a nice mesoscopic overview over the microstructure after etching (5% Nital for electrical steel) along the entire process chain. First, we can learn about the **grain shape**, how the grains become elongated in the rolling direction during rolling and again globulitic during final annealing (Figure 1a–c). Regarding electrical steel sheet, the cross-section, seen in an edge on view, is especially important in terms of a first overview because electrical steel is known for its potential microstructure and texture differences along a sheet’s thickness [17,18]. The smaller the grains’ height-to-length ratio, the higher the degree of deformation. Additionally, after hot rolling, the degree of dynamic recrystallization (small globulitic grains) can be determined and a subdivision into banded microstructure (elongated grains in rolling direction) and homogeneous microstructure (recrystallized grains) can be made (Figure 1a). Both structures have huge implications for the later recrystallization driving force (dislocation density), the texture and the cold rolling conditions.

This brings us to the second consideration, **deformation structures**. During rolling, the deformation tensor varies along the sheet thickness. Close to the surface, a shear component is present in addition to the usual compression component because of friction between the work piece and the rolls. Thus, the texture as well as the deformation structures vary across the sheet thickness [19]. After rolling without dynamic recrystallization (Figure 1b), the dislocation density is usually higher close to the surface and extremely high in narrow deformation bands (Figure 1e) as well as in shear bands (Figure 1d). Both band types form in areas with high local stresses (often close to the surface) where normal crystallographic slip cannot maintain deformation. Deformation bands are based on avalanches of crystallographic slip and, therefore, do usually not extend across grain boundaries while shear bands are not based on crystallographic slip and can intersect several grains without changing their direction [20].

Finally, the **grain size** can be measured after dynamic (hot rolling) as well as static recrystallization (hot rolling and final annealing). As mentioned above, the grain size is one of the most important factors with regard to the final magnetic properties and should always be measured, at least, after final annealing. Here it is now important, that enough grains are measured to reach statistical adequacy (>300 grains). This can usually be done easily in the sheet plane (as opposed to its cross-section). However, a distinction must be made between the different layers because the mid layer often has a different grain size from the surface area (Figure 1a,c). This, in turn, is based on the aforementioned differences in deformation structures across the sheet thickness and their implications for recrystallization. The grain size after hot rolling (including dynamic, static recrystallization and grain growth) can be important for the design of the cold rolling process to estimate process forces and because a certain degree of deformation needs to be introduced to trigger recrystallization during final annealing. In most cases, the line intercept method is used to measure the grain size according to ASTM E112-13 [21]. It should be noted that this method always underestimates the actual grain size to some degree, as the grains are cut randomly and not always along their representative extension in this two-dimensional observation.

### 3.2. Vickers Hardness (HV)

Vickers hardness measurements are a very easy and straightforward approach to give a first idea of the condition of the **microstructure**. During the test, a pyramidal diamond tip is placed on the sample’s surface for a certain amount of time with a defined force and, afterwards, the Vickers hardness number (*HV*-kgf/mm²) is calculated from the chosen force (*F*) and the area of the residual imprint of the indenter (
$\frac{2*\sin\frac{136{}^{\circ}}{2}}{d^{2}}$
, *d*-imprint diagonal) after unloading. This calculation is shown in (1), where 0.102 is a conversion factor from Newton to kilopond (see DIN EN ISO 6507 [22] for a more detailed explanation).

(1)
$$HV=\frac{0.102*2*F*sin\frac{136{}^{\circ}}{2}}{d^{2}}$$


HV is a unique unit and cannot directly be equated with pressure. The measurement tends to represent the more global **mechanical properties** of the microstructure, due to the large imprint size usually including several or many grains. If some reference HVs are known for a given material, this method can be used as a rapid tool to evaluate the microstructure after processing, as it is reflected in the hardness. For the sake of simplicity, hardness measurements are usually conducted in the surface to mid layer of the rolling direction–transverse direction plane. Here too, differences can occur along the sheet thickness because of the aforementioned reasons. Let us assume we have a Fe2.4wt.%Si sample (Table 2) with the following reference conditions: fully recrystallized with a grain size of 128 µm after final annealing at 1000 °C for 60 s-178 HV_0.2/15_ (indices: force 0.2 kg/dwell time 15 s) as well as ~50% cold rolled from a partially recrystallized 1-mm hot band-290 HV_0.2/15_. First, if we measure after a heat treatment process and determine a higher value than 178, we know that the material is probably not fully recrystallized yet. Second, if we measure a higher HV value than 290 in our sample, it follows that the driving force (dislocation density) is probably higher, which has direct consequences for the nucleus formation and growth rate.

There are also indirect mathematical interrelationships ((2) [23] and (3)) between HV and yield strength (
$\tau_{Y}$
) that can provide information about the **dislocation density** and thus the driving force during recrystallization. In these equations, *n* is the Meyers hardness coefficient, *b* the Burgers vector, *c* a constant of around 0.5, *G* the shear modulus and *ρ* the dislocation density.

(2)
$$\tau_{y}=\frac{H_{V}}{3}*0.1^{n-2}$$


(3)
$$\tau_{Y}=cGb\sqrt{\rho }$$


With well-selected reference values, the **recrystallization-start temperature,** as well as the **recrystallization fraction** and **kinetics,** can be determined well and quickly. In order to determine the recrystallization, start temperature a third reference value is needed to account for the hardness decrease through recovery. Recovery is a competing but important mechanism for recrystallization, which can already occur at lower temperatures [24]. Thereby, individual dislocations start moving by thermally activated processes (dislocation climb, cross slip, etc.) and the dislocation density (respectively recrystallization driving force or hardness) can be reduced by the annihilation of dislocations of opposite sign without recrystallization taking place. In addition, dislocations can form substructures like high-dislocation-density walls through this rearrangement, up to a point where single cells fulfil the characteristics of a recrystallization nucleus (high-angle grain boundary as an interface and low dislocation density within the cell). This phenomenon is therefore also related to the incubation time before recrystallization [20]. Unlike during recrystallization, the mesoscopic microstructure is not changed during recovery.

Figure 2 shows the use of hardness measurements to determine the recrystallization-start temperature of the already-mentioned Fe2.4wt.%Si (Table 2) steel after a total cold rolling reduction of 79.2% and of Fe3.2wt.%Si (Table 2) after a total cold rolling reduction of 50%. The microstructure of the hot-rolled state was more or less the same for both sheets (small equiaxed grains near the surface and a banded structure in the mid layer). In order to obtain the recovery reference area (mean ± standard deviation), one sample each was heat-treated at 500 °C for 24 h. The temperature and time were chosen in such a way that no recrystallization could take place but most of the recovery expected in a real final annealing could. As a result, the hardness decreased by ca. 6–9% (Figure 2a—area between the grey lines) without any mesoscopic microstructural changes (Figure 2b). Using the three reference areas of the cold-rolled, recovered and fully recrystallized states, the recrystallization-start temperature can be found from isochronal heat treatments at different temperatures and Vickers hardness testing. In the example shown in Figure 2, the recrystallization-start temperature is around 662.5 °C for Fe2.4wt.% and around 675 °C for Fe3.2wt.%Si, both determined as the intersection or definite undercutting of the recovery reference minimum by the sample hardness value from the respective isochronal heat treatments. Additional insights from Figure 2a are that the microstructure is very inhomogeneous after cold rolling (large standard deviation in hardness-areas between upper (cold rolled) and lower (recovered) line pairs) compared to the fully recrystallized state (area between blue line pairs). Moreover, as expected, the material with the higher silicon content has a higher hardness in the fully recrystallized state (higher yield strength-more brittle-more difficult to process), however, this difference is not as visible in the cold-rolled state, represented by a wide overlap (grey area) between the grades. Here, an increased hardness because of the higher silicon content of Fe3.2wt.%Si could have been balanced through the higher rolling degree of Fe2.4wt.%Si. In conclusion, Fe3.2wt.%Si has a higher recrystallization-start temperature, probably because of a lower driving force for recrystallization (cold rolling reduction of 50% vs. 79.2%) and its chemical composition (related: grain boundary mobility—Section 4.7).

With more such information at hand, whereby just one parameter is changed at a time, further research questions can be addressed; for example, how the recrystallization-start temperature is affected by parameters such as alloying elements, hot rolling degree/microstructure/texture, cold rolling degree/microstructure/texture, sheet thickness etc.

### 3.3. X-ray Diffraction

In the case of electrical steel, X-ray diffraction (XRD) is mainly used for **texture** analysis. In most cases the source is, therefore, fixed; monochromatic x-rays are used, the sample is on a three-axis goniometer and the signal is collected with an area detector. The physics are based on Bragg’s law. The results presented here have been acquired using a commercial X-ray goniometer (D8 Advance, Bruker: Billerica, MA, USA) with a high-resolution area detector (512 × 512-pixel, resolution of 5 °) and an iron anode (λ = 1.94 Å) at 30 kV and 25 mA. To measure the texture of electrical steel, we commonly use a 1-mm collimator, measure every frame for 10 s and a scan an area of at least 8 mm × 8 mm. In the end, three incomplete pole figures are combined to give a complete orientation distribution function (ODF).

The texture is important since a magneto-crystalline anisotropy is present in body-centred cubic electrical steel. The cube edges <100> (easy axes) are easiest to magnetize followed by <110> (medium axes) and <111> (hard axes) [25]. This is especially important at medium to high magnetic polarization values, where the magnetization mostly increases by domain rotation [26]. Although the classification as NO steel suggests an isotropic texture, that is not, in fact, the case. Rolling produces a relatively sharp texture (Figure 3b) and because of the nature of recrystallization (nucleus formation and growth), the final texture can also not be fully random (Figure 3c) [27]. In theory, a random cube texture in the sheet plane (<100> || normal direction) would be best for a rotating application, as there is then no hard <111> axis in the magnetization plane. However, this texture is not easy to achieve by means of standard processing routes. It becomes clear that a detailed knowledge of the texture development along the process chain is crucial to recognizing correlations between rolling conditions as well as heat treatment parameters (recrystallization, grain growth) and texture development. Again, there are differences along the sheet thickness because of the aforementioned reasons (Figure 3a).

Regarding recrystallization, there is one more interesting application of XRD, namely in-situ XRD measurements, which are not yet used extensively in the literature. During such measurements, an interesting texture peak can be selected (e.g., 0° 40° 45° on the α-fibre in Figure 3b) and continuously measured while the sample is heated. As shown in Figure 3c, the aforementioned peak then dissolves during recrystallization and thus the **recrystallization kinetics** can be determined from the time-dependent intensity reduction. Other applications of XRD include phase analysis, crystal structure analysis and strain analysis.

Based on the texture results, the **A-parameter** can be calculated to grade the texture according to its anticipated magnetic performance [28,29]. The A-parameter is calculated by averaging the deviation angle between the magnetization vector (
$\vec{M}$
) and the closest <100> axis of every single orientation within the texture (Figure 3d *y*-axis). This is done for all possible orientation relations between 
$\vec{M}$
 and the sample reference frame given by the sheet plane (Figure 3d x-axis) to account for the rotating character of the final application of NO electrical steel. The lower this value the better the texture (easy <100> axes well aligned with respect to 
$\vec{M}$
). As seen in Figure 3d, the A-parameter curve of a typical recrystallization texture can have maxima between rolling direction || 
$\vec{M}$
 (*x*-axis-0°) and transverse direction || 
$\vec{M}$
 (*x*-axis-90°). This highlights that uniaxial magnetic measurements need to be viewed critically, as they do not always catch the magnetically worst-performing sheet direction.

### 3.4. Flow Curve Determination

In this section, we present exemplary methods for determining flow curves, which are relevant for deformation processes, such as hot rolling, cold rolling or blanking. Understanding the flow behaviour of materials is of great importance to improving process strategies and investigating and understanding hardening and softening mechanisms in materials. With a good knowledge of a material’s behaviour, processes such as hot rolling, cold rolling and blanking, as well as evolving material properties, can be modelled, potentially saving time and costs. Possible questions for such models could be “How much force do I need for a defined deformation at a given temperature and strain rate?”, “How great is the tool wear?”, “How do the process parameters influence the microstructure?”, etc. The last can be important in preventing failure in the final application. For example, with regard to electric motors, thin electrical steel struts in the iron core need to withstand huge centrifugal forces. Preliminary simulations based on flow curves can be used to design the production process and predict the material properties, ensuring that the component will withstand these forces. To this end, knowledge of the **flow stress** at a certain plastic strain, which defines the stress at which plastic flow is initiated in the material under a uniaxial stress state, is essential. The change in flow stress with plastic strain is usually represented in a flow curve. There are several methods to determine flow curves, for example, cylindrical and flat compression tests in bulk or stacked layer configuration, tensile tests, torsion tests, bending tests and bulge tests. When determining flow curves, the following influencing factors need to be considered: strain, temperature and strain rate [24].

For rolling processes, the compression test is commonly used due to a comparable stress state and a higher maximum strain compared to tensile tests. Reasonable flow stress values by means of compression stresses can be determined up to a true strain of approximately 1.5. Beyond this value, friction between sample and die leads to barrelling, which falsifies the stress calculations. In order to obtain the highest possible strain range, force is applied until fracture or obvious bulging appears. After eliminating the elastic part of the stress–strain curve, the plastic strain *φ* and flow stress 
$k_{f}$
 can be calculated according to the following equations [30]:
(4)
$$k_{f}=\frac{F}{A}=\frac{4Fh_{1}}{\pi h_{0}d_{0}{}^{2}}$$


(5)
$$\varphi =\left|ln\frac{h_{1}}{h_{0}}\right|$$

where *F* is the compression force, *h_1_* and *A* are the height and cross-sectional area after compression and *h_0_* and *d_0_* are the height and diameter before compression.

Exemplary results of the flow curve determination for two electrical steels are shown below. To evaluate the hot forming behaviour, isothermal bulk hot compression tests [31] were performed on a servo-hydraulic hot deformation simulator at constant strain rates of 0.1 s^−1^, 1 s^−1^ and 10 s^−1^ in the temperature range of 800 °C to 1100 °C (Figure 4a–c). In the course of these experiments, bulk samples with an initial diameter of 10 mm and an initial height of 18 mm were placed in an upsetting cup and heated in an air circulation furnace up to 1200 °C. By holding for 10 min at 1200 °C, a homogeneous temperature distribution and similar grain size was assured in all tested samples before the compression tests at the respective deformation temperatures were conducted. Figure 4a–c show representative hot flow curves of Fe2.4wt.%Si and Fe3.2wt.%Si (Table 2) after data correction to eliminate the influence of friction and heat that occurs during deformation, as described in [31].

The approach to determining cold-to-warm rolling flow curves is often a little bit different because the available material after hot rolling, with its characteristic microstructure, is much thinner than 18 mm and the process temperatures are much lower, which limit the maximum possible strain. Therefore, cylindrical stack layer compression tests [32] are used. For this method, several discs with a diameter of 10 mm are stacked (in our case 15 discs with a height of 1 mm each) and compressed using a servo-hydraulic testing machine. Figure 4d shows some exemplary cold-to-warm rolling flow curves for Fe2.4wt.%Si tested at different temperatures (lower compared to Figure 4a–c) and strain rates using the described stack layer compression tests.

In general, the flow stress increases with increasing strain rate and decreases with increasing temperatures [33]. The reduced or even inverse strain rate sensitivity at 400 °C can be explained by dynamic strain aging [34]. As long as yield or flow stress anomalies are absent, the presented data can directly be used to interpolate and model material behaviour as a function of strain rate and temperature during processing. The information can also often be used to infer underlying mechanisms of deformation based on the activation energies and volumes of the thermally activated processes, i.e., those that are affected by rate and temperature [35]. However, care must be taken not to conflate changes in or contributions from several mechanisms. For greater separation of mechanisms, smaller-scale tests may also be used, see Section 4.1. As expected, a comparison of different compositions (Figure 4a–c) reveals the effect of silicon content on the mechanical properties; here, a higher flow stress for Fe3.2wt.%Si compared to Fe2.4wt.%Si was observed in all test conditions.

The shape of the flow curve provides information about the material’s dynamic hardening and softening behaviours [24,36]. A continuous increase in flow stress followed by a flat maximum and a decreasing stress is usually a sign of strain hardening followed by recovery. However, the development of more pronounced maxima, in extreme cases even a periodically rising and falling of the flow stress as straining proceeds, reveals the occurrence of dynamic recrystallization that is concurrent recrystallization and deformation instead of static recrystallization during a heat treatment after deformation. Here, the flow curves in Figure 4a–c reveal that Fe3.2wt.%Si tends to show more dynamic recrystallization at a temperature of 800 °C and a strain rate of 1 and 10 s^−1^. All other hot compression testing conditions reveal a flow curve shape that represents recovery. Recovery is a typical softening mechanism at elevated temperatures for high-stacking-fault materials such as electrical steel, where the absence of dislocation dissociation or small stacking-fault widths allows easy thermally activated rearrangement of dislocations by climb or cross-slip [24].

In order to model flow curves, various mathematical approaches can be used. An example is the semi-empirical flow curve model (6) proposed by Hensel and Spittel [37] that is fitted to the presented experimental data in order to be able to model the interpolated flow stresses *k*_f_ as a function of true strain *φ*, strain rate 
$\dot{\varphi }$
 and temperature *T* in °C.

(6)
$$k_{f}=A\;\cdot \;e^{-m_{1}\;T}\cdot T^{m_{9}}\cdot \;\varphi^{m_{2}}\cdot e^{m_{4}/\varphi }\cdot {\left(1+\varphi \right)}^{m_{5}\cdot T}\cdot e^{m_{7}\cdot \varphi }\cdot {\dot{\varphi }}^{m_{3}}\;{\dot{\varphi }}^{m_{8}\cdot T}$$


The determined parameters for *A* and *m_i_* for Fe2.4wt.%Si and Fe3.2wt.%Si based on the hot flow curves shown in Figure 4a–c are listed in Table 3.

### 3.5. Magnetic Properties

Various standardised measurement set-ups can be used for the metrological **magnetic characterization** of electrical sheets. Thereby, single-sheet testers and Epstein frames are the most common sensors. Both these sensors work according to the same measurement principle: when a magnetic field is introduced and thus a magnetic flux is generated in a sample, a voltage is induced in a secondary copper winding that is proportional to the magnetic polarization of the sample. The magnetic field is generated by a primary winding. In case of standardised sensors, as presented in Figure 5, both the secondary and primary winding are placed around the sample with the secondary winding being closest to the sample. The frequency and polarization amplitude are generally sinusoidal for standardised Epstein frame and single-sheet tester measurements. The differences between the two setups stem from the sample geometry and magnetic flux path. In an Epstein frame four strips are positioned in a rectangular shape with overlapping edges. The primary and secondary coils are placed around each of the four limbs (Figure 5a). The magnetic flux path is closed solely by the magnetic core material that is to be characterized. With a single-sheet tester on the other hand, the coil is placed on one sheet and a double c-yoke closes the magnetic flux path. The advantage of the latter setup is a homogeneous magnetic field distribution along the sample cross-section. In an Epstein frame the magnetic path length is different in the outer and inner corners of the rectangle and thus, especially shortly before approaching saturation, a flux concentration at the inner diameter occurs. Therefore, an inhomogeneous magnetic flux distribution is found along the sample cross-section. The advantage of an Epstein frame over a single-sheet tester, on the contrary, is the non-present distorting effect of the yoke. With the Epstein frame, no additional losses can occur. With the single-sheet tester, the yoke can lead to additional losses. Moreover, the single-sheet tester measurements are limited to one preselected sample direction. With the Epstein frame, both the rolling and transverse direction can be measured at the same time and a mean value can be obtained. The sample sizes can be as large as 500 mm × 500 mm for single-sheet testers and 280 mm × 30 mm for Epstein frames in order to have enough sample volume to account for inhomogeneities. We refer to DIN EN 60404 [38] for further information.

Depending on the experimental design, electrical steel can be analysed thoroughly on standardised measurement setups. For example, in Figure 6 the effect of silicon content on the magnetic properties has been measured. The electric steel samples considered had different silicon contents of 2.4 wt.% and 3.2 wt.% (Table 2), a sheet thickness of 0.5 mm, a standard texture (comparable to Figure 3c) and a similar grain size of 224 µm and 231 µm, respectively. The magnetization behaviour is distinctly better for the electrical steel with the lower silicon content when approaching saturation, as the higher proportion of ferromagnetic iron leads to a lower electric resistivity (Figure 6a). In terms of magnetic loss, an increased silicon content is generally favourable. However, at low frequencies (50 Hz, Figure 6b), the magnetic losses are still almost the same. This is due to the comparable grain sizes and their strong influence on the hysteresis loss component, which is dominant at such low frequencies (Table 1). If the magnetic properties are measured at different frequencies, e.g., at 1000 Hz (Figure 6c), the effect on the classical loss component can be isolated. The increased silicon content leads to a higher electrical resistivity and thus to lower classical loss.

With an extended choice of experimental excitation parameters, besides the classification DIN norm point at 1.5 T and 50 Hz [38], i.e., a higher field strength range and various frequencies or other excitation wave forms, electrical steel can be characterized in great detail on standardised setups, however, all these setups have systematic simplifications and ideal measurements condition so that advanced characterization methods (see Section 4.3 and Section 4.4) are still needed to improve the understanding of electrical steel.

## 4. Advanced Methods

There are still some open topics in the literature, which are very important in view of electrical steel. In particular, a better understanding of the fundamental phenomena underlying the properties and processing steps discussed above will help to formulate physically based models for process modelling and optimization, eventually enabling the development of new material and process design strategies [6].

One big topic is recrystallization. During recrystallization, the residual stresses are minimised, the final texture develops and the grain size is adjusted. All three factors are crucial for the final magnetic properties. However, little is known about the nucleus formation during rolling and recovery or about the mobility of specific grain boundaries during recrystallization. Consequently, few predictions can be made based on the initial microstructure and texture about the recrystallization kinetics as well as about the resulting microstructure and texture. Another topic is the local magnetic properties and their correlation to domain structures, specific grain boundaries and deformation structures. In this section, different analysis methods are proposed to gain insight into these topics.

### 4.1. Nano- to Micromechanics

Sheet thicknesses of NO electrical steel sheet have been trending downwards to reduce classical loss, to the point where they are now approaching the grain size of the steel. Hence, the final blanking process is influenced by the anisotropic deformation behaviour of single grains and individual grain boundaries. The question now is how to investigate, understand and include this single and bi-crystal level with respect to the **deformation behaviour** and evolving **deformation structures**? In the past, suitable compression experiments of iron silicon single and bi-crystals have been carried out on the macroscale [39,40], however, because of the difficult and laborious sample manufacturing, only few macroscopic single and bi-crystals have been tested so far. By switching to the microscale, a much larger number of single and bi-crystals, and therefore the grain boundaries they contain, can be investigated in terms of their mechanical properties starting from a polycrystalline sheet with big grains. Two approaches to do this can already be found in literature, namely nanoindentation [41] and micropillar compression [42].

The principle of **nanoindentation** or instrumented indentation is comparable to Vickers hardness measurements (Section 3.2), only on a much smaller scale with indentation depths from a tens of nanometres to a few microns [41]. As a result of this smaller scale, the area of the imprint is not measured directly by microscopy but is inferred indirectly from the load-displacement data recorded, as well as from the knowledge of the shape of the indenter tip. Usually, a diamond Berkovich tip is used, as it provides the same cross-sectional area as a function of depth as the Vickers pyramid but, owing to its three instead of four faces, it is easier to achieve a single sharp apex during its production, which is essential for indentation at small depths. Starting from the measured load and depth after surface contact, multiple values can be calculated, such as hardness, indentation modulus and rate or temperature dependence of flow, and can be correlated to different phenomena with a high degree of precision, taking the elastic deformation of the tip as well as of the bulk material into account (Figure 7a grey curve—standard nanoindentation curve). For a more detailed explanation of the method and it application to derive rate- and temperature-dependent properties, we refer to [41,43,44]. In measurements on very carefully prepared samples without mechanical induced defects close to the surface, even phenomena related to the nucleation of dislocations in the material’s volume or during transmission of flow through grain boundaries can be investigated. For example, different studies have shown that a second pop-in, as opposed to the first pop-in that is related to the transition from the elastic to the plastic regime and thus to the activation of dislocation, can be seen in load-displacement curves when an indent is placed close to a grain boundary that acts as an obstacle for moving dislocations [45,46]. Following the initial elastic, Hertzian loading and pop-in, which signifies dislocation nucleation (both may be absent if dislocations are already present in the loaded volume), the slope of the load-displacement curve naturally increases due to the increasing cross-section of the indenter in contact and hardening of the material within the plastic zone around the tip. The latter is associated with (immobile) dislocations piling-up at and exerting increasing stresses on obstacles like grain boundaries. At some point, the dislocations are either transmitted or new sources are activated across the grain boundary and plasticity can continue with an avalanche of dislocations being able to move again (Figure 7a, blue curve—displacement burst or second pop-in). The load at the second pop-in in combination with the indent’s distance to the grain boundary can then be used to calculate how strongly the grain boundary acts as an obstacle and how this, for example, depends on the grain boundary characteristics (rotation axis, rotation angle, inclination etc.). Therefore, this method allows us to learn about the role of this specific grain boundary during deformation. If indents are placed far away from grain boundaries, the method can equally be used to investigate the orientation dependent mechanical properties of single crystals. Another approach is to place an array of indents on a cross-section of a shear cut sheet close to the cutting edge (Figure 7b). The hardness variations can then be correlated to residual stresses that remain after blanking, where both the magnitude and the depth of these stresses are interesting. Residual stresses can, in turn, be correlated to dislocation densities, which deteriorate the magnetic properties.

The biggest drawback of nanoindentation is the complex stress state underneath the indenter tip, which makes it difficult to understand the activation of individual slip systems. This, on the other hand, is especially important in order to learn more about the influence of grain boundaries on the activation and movement of dislocations. In this context, **micropillar compression** tests can be a useful tool, because, here, a more uniaxial stress state is present [48]. Micropillars can be milled site-specifically with a focused ion beam (FIB) in a dual beam scanning electron microscope. Standard diameter dimensions of cylindrical micropillars are 0.5–8 µm with a diameter-to-height ratio of 1:2.5. Site-specificallymeans that preselected single orientations from an EBSD measurement or specific grain boundaries can be tested within a polycrystalline material with large grains. It is even possible to carefully control the share of the respective grains in a bi-crystalline micropillar with a specific grain boundary [49,50]. After compression, the active slip systems can then be analysed through different methods such as visual slip trace analysis, plane tilt analysis or cross-section EBSD (Figure 8).

In a next step, the found slip systems can be correlated to the stress–strain curves and, with the help of Schmid factors, critical resolved shear stresses can be calculated, or the extent of non-Schmid flow can be quantified. This information can be used to inform crystal plasticity based deformation models [6], although size effects have to be accounted for [52]. Furthermore, the dislocation structure within a small deformed single crystal or around the boundary of a bi-crystalline micropillar can be analysed by transmission electron microscopy after preparing a thin lamella out of the micropillar. In the case of electron-transparent lamellae from nanoindents or micropillars, the area of interest will always be highly relevant because the dimensions of the plastic zone (nanoindent) or the compression sample itself are already in the order of magnitude of the lamella, which is otherwise often a problem in the high-resolution characterization of macroscopic samples. Today, it is already possible to do these compression tests in-situ, whereby visual events, for example at a grain boundary, can be directly correlated to events in the stress–strain curve [53].

In order to record information from nanomechanical testing that is relevant to the actual blanking process and its high strain rates, micropillar compression tests and nanoindentation can also be performed under extreme conditions. This includes micropillar compression tests at strain rates of up to 100 s^−1^ and nanoindentation or nanoimpact tests with up to 1000 s^−1^ [54]. Similarly, the test temperature can be varied from a hundred or so degrees below room temperature [55] up nearly 1000 °C above it [56] to explore the temperature dependence of plasticity.

Of course, it is important to recognise where a reduction in sample dimensions leads to quantitative or qualitative size effects. In the context of bi-crystal deformation, which is essential to the blanking of thin sheets, Heller et al. [49] have already shown, using microscopic and macroscopic single and bi-crystals of Fe2.4wt.%Si, that the deformation behaviour at grain boundaries on the microscale can be directly related to that on the macroscale.

### 4.2. Crystal Growth

One disadvantage of the experiments on the microscale is that the deformation structures cannot be directly correlated to the magnetic properties, since it is not possible to measure these on such a small scale with standardised methods. Moreover, the mesoscopic domain structure, controlling the magnetic properties, on such a small scale is not comparable to the one on the macroscale, as the domain size itself is in the same order of magnitude as the micron-sized compression samples. Therefore, specific interesting single, bi- and oligo-crystals can be grown on the macroscale, followed by macroscopic uniaxial compression tests, which are comparable to the aforementioned micropillar compression tests. While they allow the analysis of evolving deformation structures using similar techniques to those applied in case of nanomechanical testing (scanning electron microscopy (SEM), electron backscatter diffraction (EBSD), electron channelling contrast imaging (ECCI), transmission electron microscopy (TEM) etc.), larger samples additionally allow the correlated measurement of magnetic properties, as for example described in Section 4.3. In general, the challenges during crystal growth include the prevention of unwanted nucleus formation of additional grains at the crucible wall, finding stable growth directions of the respective crystal structure and, for bi-crystals, keeping the grain boundary as straight as possible. For electrical steel, the high temperatures required to melt the material during the process pose an additional challenge.

There are various techniques to grow a crystal with a preselected orientation. Here, we will focus on the vertical Bridgman–Stockbarger method [57]. A custom-built induction furnace for this method is shown in Figure 9a. For the preparation of iron-silicon single and bi-crystals, a crucible made from Al_2_O_3_ is first fixed in a glass cylinder with boron nitride. Then, the steel is inserted as a blank, which is polycrystalline, with one (for a single crystal) or two (for a bi-crystal) nucle(us)i attached to the bottom of the blank (Figure 9b). Thereby, the respective nuclei are single crystalline with preselected orientations obtained from previous crystal growth or a very large-grained sheet. Typical process conditions to prepare electrical steel samples with dimensions of the order of 1.5 cm and several cm in length, include a maximum current output of 260 A and a growth rate of 0.12 mm/min, with a rotation during growth of 6 rpm. The latter is important since the induction field is not perfectly homogeneous with regard to the crystal blank, thus rotation can minimise unwanted nucleation.

In previous studies, some single and bi-crystals have already been investigated for their magnetic properties, giving first insights on the magneto-crystalline anisotropy [58], the role of deformation [39] and the influence of grain boundaries [59]. However, literature lacks a systematic study which correlates evolving deformation structures with specific grain boundaries and their implications for the final magnetic properties. Moreover, most of those studies are quite old and, in the meantime, new characterization methods, such as electron channelling contrast imaging (ECCI) or high-resolution electron backscatter diffraction (HR-EBSD), have been developed and can now resolve deformation structures much better.

### 4.3. Miniature Single-Sheet Tester (miniSST)

In order to correlate the effects of deformation structures in single crystals and at grain boundaries with the magnetic properties in a direct way, a correlative method is required that allows both systematic deformation and magnetic characterization experiments on the same sample. We therefore outline, here, a new approach, which is being developed to fundamentally investigate the interaction of grain boundaries and dislocations and its effect on the magnetic properties in thin sheets. This approach has the potential to explain the deterioration of magnetic properties during the blanking of thin sheets, especially where the sheet thickness is approaching the grain size.

To this end, the single-sheet tester, already presented above as a standard tool for the magnetic characterization of sheet material (Section 3.5), has been miniaturised to study the **magnetic properties** of single, bi- or oligo-crystals with dimensions achievable by regular crystal growth, as described in Section 4.2. Since the sample sizes resulting from crystal growth are in the order of 2 cm², a correspondingly small single-sheet tester had to be developed. The outer distance between the magnet yoke legs of the newly designed (in cooperation with Dr. Brockhaus Messtechnik GmbH & Co. KG: Lüdenscheid, Germany) miniature single-sheet tester is 22.3 mm (Figure 10a). The resulting minimum sample size is 25 mm × 10 mm (length × width; length can be larger, width is fixed). The primary and secondary winding system has 60 turns each. The single-sheet tester is controlled by an MPG 200 unit with MPF-Expert software (both from Dr. Brockhaus Messtechnik GmbH & Co. KG). Within this framework, a numerical air flow compensation, developed at the Institute of Electrical Machines, RWTH Aachen University (Aachen, Germany) is used.

With this method, the influence of grain boundaries, crystal orientation or dislocation density on the magnetic properties of iron silicon electrical steel can be quantified in a systematic manner and different crystal orientations, grain boundary types and deformation conditions can be explored in isolation.

Figure 10b shows the polarization curves of a grain-oriented electrical steel (Goss texture) magnetized in three different directions relative to the rolling direction at 50 Hz. Due to the coarse grain structure in combination with the small sample size, it is possible to analyse only a few or even only one single grain and its orientation impact. Furthermore, the results can be used as a proof of concept for the miniature single-sheet tester. The magnetization behaviour shows the expected strong crystallographic texture effect of grain-oriented electrical steel (0°~easy <100> axes || magnetization direction vs. 90°~medium <110> axes || magnetization direction vs. 45°~hard <111> axes || magnetization direction, see Section 3.3). In the high-polarization region, the transverse direction (TD) is not equal to the hardest magnetization direction, but, due to geometric relations, the 45° direction is closest to the hardest theoretical magnetization direction (<111> || magnetization direction), which should be found at a ~55° rotation around the sheet normal. The behaviour at lower polarizations can be explained by the residual stress state after the rolling process. This is composed of a small tensile stress in the rolling direction and a small compressive stress in the transverse direction, making TD the hardest magnetization direction in the domain growth regime, as compressive stresses heavily deteriorate the magnetic properties [60].

### 4.4. Vector Hysteresis Sensor

Although commonly used, single-sheet testers and Epstein frames are based on one strong simplification: unidirectional properties and scalar values of the magnetic field and the magnetic polarization. However, in reality the magnetic field and flux are vectors that depend on the domain structure of the material and become collinear in the case of saturation, when the flux is forced in the direction of the external field. The phase shift between the magnetic polarization and magnetic field depends on the material, e.g., crystallographic texture and residual stress state as well as measurement conditions, e.g., applied mechanical force. A vector hysteresis sensor can measure these **vector properties** by means of four needles that pick up the magnetic flux and H-coils to determine the magnetic field [61]. The exemplary sensor displayed in Figure 11a was built by Dr. Brockhaus Messtechnik GmbH und Co. KG. In order to magnetize the sample a yoke needs to be magnetized. The setup displayed in Figure 11b was designed at the Institute of Electrical Machines, RWTH Aachen University. Therein, strip samples can be installed in a universal tensile testing machine in order to apply a tensile stress while generating a magnetic field that goes through the sample by means of the yoke. Furthermore, the yoke can be rotated to investigate different directions of mechanical stress in relation to the direction of magnetization. The overall setup to obtain vector properties is more complex and the measured volume is significantly smaller (12 mm × 12 mm), compared to a standard single-sheet tester (60 mm × 60 mm); however, it has the potential to provide additional insights into stress state-dependent vector properties that are currently not well understood.

### 4.5. Bitter Imaging

During magnetization, domains grow through the movement of domain walls and rotate in the final stages of polarization according to the magnetization vector. Thus, a detailed knowledge of the domain structure can help in understanding the magnetization behaviour in more detail. One mesoscopic technique to indirectly visualise the **domain structure** is the Bitter imaging technique [63]. In contrast to other methods requiring special equipment, such as Kerr microscopy at the microscale and magnetic force microscopy at the nanoscale, Bitter imaging can be performed using a conventional optical microscope. The basic principle is based on the accumulation of Fe_3_O_4_ particles along horizontal surface stray fields, which result from the domain structure underneath the surface. In this context, it must be considered that the free surface alters the domain structure and therefore bulk information can only be extracted indirectly [64]. Here, colloidal Fe_3_O_4_ nanoparticles are used, a so-called ferrofluid. The particle size determines the resolution [65]. After dripping and distributing some ferrofluid on a weakly magnetized surface, the forming domain pattern can be analysed with a light microscope, revealing the surface domain structure (Figure 12). Most of the pioneering work regarding this technique was done more than 80 years ago [63] and, in current research, the highly advanced methods like Kerr or magnetic force microscopy are often used in specialised research on magnetic materials. Of course, no single technique can deliver all the required results. While magnetic force microscopy is limited to small scales, Kerr and Bitter imaging both reveal domain structures at the microscale. However, Hubert et al. found that some domain structures can only be visualised by Bitter imaging, while others may only be visualised by Kerr microscopy [64].

In the case of electrical steel, there is a demand for interdisciplinary research, from the processing methods (materials engineering) over the deformation and microstructure evolution processes (materials physics) to the evaluation of the magnetic properties of sheets in final geometry (electrical and mechanical engineering). For this, a simple method, applicable in most materials science laboratories, is favourable. As such, the method described here can easily find application to address the outstanding issues outlined above, namely to reveal magnetic domain structures in a systematic study of single, bi- and oligo-crystals (Section 4.2), as well as different deformation states. Combined with the measurement of the magnetic properties (Section 4.3), the imaging of the deformation-induced changes in domain structure can help researchers to better understand the interaction of deformation-induced defects, grain boundaries, crystal orientation and iron loss. First investigations in this direction showed different patterns for different orientations, as well as different deformation states and a correlation between grain boundaries and domain walls (Figure 12). Moreover, transition zones towards grain boundaries and a finer domain substructure (inlet Figure 12) in the mesoscopic domain structure are often observed.

### 4.6. Neutron Grating Interferometry

For measuring global magnetic properties of electrical steel mostly single-sheet testers or Epstein frames are used (Section 3.5). However, these machines cannot resolve local magnetic properties. In this regard, neutron imaging can be a great tool, thanks to its high penetration depth in metals and its interaction with magnetic fields. The neutron-based measurement method, providing the highest quality, is called neutron grating interferometry and was first described in 2008 [66]. To realise neutron grating interferometry, a composition of three gratings (source grating, phase grating and analyser-attenuation grating) as well as a wavelength selector need to be introduced into the beamline. With this system ultra-small-angle neutron scattering can be visualised on a detector screen [67]. Ultra-small-angle neutron scattering is based on the fact that neutrons are refracted whenever they travel through an interface that separates materials of different refractive indices. In electrical steel, domain walls separate areas of varying magnetization. Since the refractive index correlates with the magnetization, neutrons are scattered at such domain walls. The resulting image on the screen is called a dark field image (DFI). It contains information about the **domain wall density** in the volume of the sample that was radiated through. A low DFI-signal on the detector screen indicates much scattering and, thus, small magnetic domains [68].

It is well established in the literature that stresses have a direct impact on the magnetic properties. For example, compressive stresses generally increase the magnetic losses. Thereby, excess loss is more affected than hysteresis loss and classical loss is hardly influenced. Moreover, the effect is stronger at low rotation frequencies and polarizations. The situation is somewhat different for tensile stresses, it was found that magnetic losses first slightly decrease at low tensile stresses before returning to the initial level [69,70]. Residual tensile and compressive stresses are, at present, unavoidable in the final part. For example, blanking, which is a fundamental production step for most electromagnetic parts such as rotors and stators, introduces unwanted residual stresses in the area next to the cutting edges. Much work has already been done in order to reduce blanking related residual stress [47,71,72]. Beyond that, after the electrical steel sheets have been stacked and fixed in the stator or rotor, magnetostriction and/or centrifugal forces can lead to additional stresses. However, recent publications have shown a possible application for this stress-induced deterioration of magnetic properties. According to several authors [73,74,75], embossings can be used instead of cut-outs to generate the necessary flux barriers in reluctance machines, whereby the mechanical strength of the rotor or stator could be increased. It has been shown that this deterioration of magnetic properties is based on a reduced domain size correlating with an elevated domain wall density, which, in turn, can be evaluated by neutron grating interferometry [75,76].

Weiss et al. [76] successfully applied neutron grating interferometry to evaluate the influence of silicon content, cutting clearance, sheet thickness and magnetization frequency on the magnetic domain structure of electrical steel sheet samples. Figure 13 shows the DFI contrast (C_DFI_) for a wire-cut and an exemplary punched sample. Both samples are cut perpendicular to the rolling direction, have a silicon content of 2.30 wt.% and a sheet thickness of 0.35 mm. A worn tool and a cutting clearance of 35 µm were used for the blanking process. Both samples were magnetized at a magnetic field strength of 780 A/m during the measurement. In this case, it could be shown, by the help of neutron grating interferometry, that the stress affected area reached roughly 1 mm deep into the punched sample and that the domain size was reduced near the cutting edges. In comparison, the domain structure of the stress-free wire-cut sample was homogenous throughout the whole sample.

Gilch et al. [75] used neutron grating interferometry to evaluate the magnetic domain structure next to embossings in electrical steel sheets having a thickness of 0.35 mm and a silicon content of 2.4 wt.%. A four-sided pyramidal geometry with a tip angle of 136.5° in combination with a flat die was used for embossing. The embossing force was varied between 50 N, 100 N, 200 N and 400 N. For each sample, twenty embossings were generated every 0.5 mm along the rolling direction of the sample. Figure 14 shows the neutron grating interferometry signals for two different samples that were embossed with (a) 100 N and (b) 200 N. During the measurement, both samples were magnetized at 813 A/m perpendicular to the line of the embossings. The DFI signal was normalised to a stress-free reference sample. In Figure 14a,b the respective areas around the embossings show a lower DFI signal (darker areas along z = 0 mm). For the higher embossing force of 200 N, the signal is generally lower and the affected material volume is larger. Therefore, the authors were able to demonstrate that embossing affects the local magnetic domain structure in electrical steel samples. By comparing neutron grating interferometry results with a mechanical finite element analysis of the embossing process it was further shown that areas of especially high domain wall densities coincide with areas of high compressive stresses.

A third interesting application of neutron grating interferometry could be the analysis of deformed and un-deformed single, bi- and oligo-crystals which were already introduced in Section 4.2. In this way, the local domain structures found could be correlated to local dislocation structures at specific grain boundaries and the overall magnetic properties of the sample (Section 4.3).

In summary, neutron grating interferometry offers the possibility of locally evaluating the magnetic domain structure of electrical steel sheets through their volume, which is a major advantage compared with global magnetic measurements (Section 3.5 and Section 4.3) and the imaging of surface domain structures only (Section 4.5) and will potentially contribute to the optimization of electric drive layouts and designs in the future.

### 4.7. Recrystallization Scratch Experiments

Recrystallization during final annealing is one of the most important phenomena for electrical steel since the dislocation density is heavily reduced, the final texture develops, and the final grain size is adjusted. All three are very important factors for the final magnetic properties. Recrystallization is composed of nucleus formation and nucleus growth. Until now, how and where the nuclei form remains under discussion in the literature. It is, however, known that only nuclei of a critical size with mobile high-angle grain boundaries and a low dislocation density relative to the surrounding volume can grow. Such nuclei can form during deformation or recovery by the formation of substructures through the accumulation of dislocations. This means, conversely, that the nuclei do not form randomly or originate anew; they are already pre-existing before recrystallization and thus have orientations that evolve during rolling [27]. In order to be able to model recrystallization and grain growth on a physical basis, the nucleus formation process as well as the grain boundary mobility (grain growth) and the factors influencing both need to be understood. The following characterization methods, recrystallization scratch experiments and quasi-in situ EBSD (Section 4.7 and Section 4.8) provide approaches to understanding these aspects better.

With the help of the first approach, the **grain boundary mobility** can be determined. Figure 15 shows a schematic layout of the experiment first proposed by Basu et al. [77].

First, a single crystal needs to be grown (Section 4.2). Next, a cuboid must be cut out of the single crystal, which is subsequently cold rolled to introduce a driving force (new dislocations) for recrystallization. For this, it is important that the single crystal’s orientation and the resulting ductility’s temperature dependence are taken into account because some orientations are so brittle that they fracture during rolling at room temperature before the driving force is high enough. A minimum degree of deformation of 80% is recommended. Below this value, recrystallization might not occur; however, this will of course strongly dependent on the material. Since there are no high-angle grain boundaries and possibly no shear or deformation bands in the single crystalline sheet, nucleation sites need to be introduced in addition to the cold rolling procedure. Bringing in a defined scratch on the surface after initial metallographic preparation results in nicely localised nuclei, which can be encouraged to grow during heat treatment. Good results for the scratch can be readily achieved in a well-equipped mechanical workshop; here, we simply dragged a hard metal lace fixed in a CNC machine across the sample at a depth of 0.06 mm and a speed of 13 m/min. Afterwards, the sample needs to be fully metallographically prepared for EBSD without grinding away the scratch. In order to derive the resulting driving force for recrystallization in the surrounding, cold-rolled volume, which is based on the dislocation density and is an important factor for the later mobility calculation, different techniques can come into play. There are indirect techniques, such as Vickers hardness (Section 3.2) or the measurement of geometrically necessary dislocation density by EBSD, as well as direct techniques to image dislocations, such as ECCI or TEM. For the former technique, Equations (2) and (3) from Section 3.2 must be combined and inserted into (7) in order to calculate the driving force (*F_dd_*-driving force based on the dislocation density (*ρ*)) [20]:
(7)
$$F_{dd}=\alpha *G*b^{2}*\rho$$


After determining the driving force, the sample is ready for heat treatment. For this, it is important to use a reducing annealing atmosphere, e.g., forming gas, to prevent the prepared surface from oxidation, and the heat treatment parameters must be chosen carefully in order to trigger growth at the scratch but prevent nucleus formation in the matrix. Results from one successful recrystallization scratch experiment can be found in Figure 16. In the upper third of (a) the single-coloured matrix (purple) can be seen containing some substructures (dark lines); in the lower third, the non-indexed scratch (black) is visible and, in the middle, nuclei grow from the scratch into the matrix. Through combining the free growth length of the nuclei *r*_gl_, the nuclei orientation, the heat treatment parameters (temperature 765 °C and time *t =* 300 s) and the driving force *F_dd_* (calculated based on HV_0.2/15_ = 303.9), the mobilities of grain boundaries, classified according to their misorientation angle *g,* can be calculated (Figure 16b). Grain boundary misorientations are only measured between nuclei and the matrix.

These first results of an Fe2.4wt.%Si sample (Table 2) are about two orders of magnitude below the results of the investigations of Wits et al. [78], who studied the body-centred cubic/face-centred cubic interface mobility in iron, and of Kim et al. [79] who used the aforementioned results for the grain boundary mobility during texture development investigations in body-centred cubic iron sheets. The reasons for the lower measured mobility could be surface effects, a general influence of the alloying elements (Si, Al, Mn etc.), solute-drag (C, N etc.), an overestimation of the driving force based on Vickers Hardness (~26 MPa), or the mobility is overestimated in the few studies available for this material. However, the qualitative shape of the distribution in Figure 16b, with very low mobilities for low angle grain boundaries (0–15°), a high point between 30–50° and a steep descent towards 90° is typical for cubic metals [20,80]. Future analyses could focus more on the role of the rotation axis or special grain boundaries in body-centred cubic materials. The parameters, which can and should be varied in this kind of experiment to learn more about the influencing factors, are: the alloy concept (and thereby the extent of segregation to grain boundaries and solute drag [81]), the degree of deformation and heat treatment parameters. In this way, the method provides a viable route to collect grain boundary mobility data for many boundaries at the same time and allows an efficient comparison of different alloys or processing conditions, as is urgently needed for the physical modelling of microstructure evolution during the heat treatment of rolled, polycrystalline electrical sheets.

### 4.8. EBSD and Quasi-In-Situ EBSD

The second approach targets a detailed characterization of **deformation structures** as well as the **nucleus formation** and **growth** behaviour during recovery and recrystallization. Below, an exemplary experimental procedure is described based on a Fe2.4wt.%Si steel sheet (Table 2) that has been cold rolled ~50% from a partially recrystallized 1-mm hot band. First, electrical steel sheet samples are prepared as cross-sections with a final colloidal polish (0.04 µm OP-U suspension). Electro polishing is not possible in this case, since it etches away the corners and edges, which are particularly interesting, as they include the surface area with the highest shear deformation (most nucleation sites). In the next step, an EBSD panorama map is measured along the edge surface resulting in a large map—here, 6.5 mm × 0.5 mm in size for a sheet sample measuring 12 mm × 10 mm × 0.5 mm. This size is important in order to analyse a representative area. Close-up maps for more detailed microstructure information (shear bands, deformation bands, nuclei, substructures) can be added afterwards. After this initial characterization of the cold rolling state, the sample has to be annealed. Temperature and time must be selected carefully to trigger mainly nucleus formation and not too much growth. Moreover, annealing should be carried out again under a reducing atmosphere in order to prevent oxidation, as the sample cannot be re-prepared because we want to directly correlate the microstructure after annealing to the initial cold rolling microstructure, which is only possible by staying within the same edge or surface layer. Now, the EBSD panorama and close-up maps are repeated along the same areas as before. This annealing–imaging sequence can be repeated as long as the sample surface remains sufficiently pristine for EBSD measurements.

From the first map (Figure 17a), we can learn a lot about the initial cold rolling microstructure, however, the higher the degree of deformation the lower the quality of the EBSD patterns will become. For instance, an indexation rate of above 95% was achieved for a cold rolling reduction of 50%, but just 46% were indexed for a sample with a cold rolling reduction of 82.5%. Through an analysis of the geometrical necessary dislocation (GND) distribution, the grain orientation spread (GOS) distribution or the small-angle grain boundary distribution, conclusions about the dislocation distribution and thus about the stored elastic energy after deformation can be drawn. It has been shown, for example, that more energy is stored in γ-fibre components than in α-fibre components [82]. Moreover, the GND density can be used to estimate the dislocation density and thus the driving force for recrystallization [83]. A final point is that shear and deformation bands can be better characterized than with light microscopy or secondary electron images, due to the combination of a large area, sufficient resolution and crystal orientation information.

If the heat treatment parameters were chosen right, first nuclei will be present in the second EBSD-map recorded after a first heat treatment. These nuclei can be isolated, for example, through a GOS filter (<1.5 °), and saved in a separate partition (Figure 17b). Afterwards, specific nucleus orientations can be marked in this partition. In a final step, the marked coordinates can be found again in the initial cold rolling map. Through this procedure, specific nuclei can be directly correlated to the initial local cold rolling structure. This knowledge, in turn, can help in understanding the necessary characteristics of nucleation sites in more detail and, if more than one annealing step is reviewed via EBSD, the understanding of nucleus growth characteristics can also be improved.

Beyond EBSD orientation analysis, electron channelling contrast imaging (ECCI) [84], as a distantly related technique, can be a great tool to directly reveal the local dislocation structures [85], especially around potential nuclei or after nanomechanical testing (Section 4.1). With this technique, the orientation of a grain is first carefully characterized through EBSD. In a next step, the edge of a Kikuchi band with a high contrast is chosen for good channelling conditions. Through rotating and tilting the stage, this band needs to be moved into the centre of the electron beam. Now most electrons will move deeply into the material and the backscattered signal is very low. Only around imperfections, like dislocations, are the electrons scattered and can be collected with a backscatter electron (BSE) detector, giving a bright signal on a dark background for crystal defects. Combining the visual trace and visibility criteria of a dislocation with the Kikuchi pattern data, a lot of information can be obtained regarding the overall dislocation structure, as well as the dislocation type. Besides, since the dislocations actually become visible, they can be counted and a quantitative dislocation density can be measured more easily than by transmission electron microscopy with its very limited volumes and the need to thin the sample to the order of 100 nm (important, for example, in estimating the driving forces for recrystallization, see Section 4.7). [86]

## 5. Discussion

### 5.1. Evaluation of the Characterization Methods

Manufacturers of NO electrical steel sheet usually specify the following sheet parameters: sheet thickness, maximum iron loss in transverse and longitudinal direction at 50 Hz and 1.5 T, minimum magnetic polarization to reach several magnetic flux densities, yield strength, Vickers hardness and density [8]. These parameters can give a good first idea of the respective material, however, they are not detailed enough to select a material specifically for an application. For example, how does the material behave at higher frequencies or are there harder magnetization directions in between rolling and transverse direction that are systematically neglected by standardised testing methods? To answer this question, knowledge about the grain size as well as the texture (A-parameter) and alloying concept would be very helpful. This knowledge and more can be gained with the help of the characterization methods described above.

Table 4 summarises all characterization methods described here. From left to right the name of the method is given, it is classified by the process step to which the characterization method is relevant, the effort is evaluated, the potential insight is described and some additional information is provided in a comment column. Moreover, the colour code indicates how broadly the respective method should be used in everyday business. Methods with a good effort-to-insight ratio (green), such as light microscopy or hardness measurements, should always be used, be it with regard to quality management or in R & D, in order to benchmark and improve a process or material. Methods highlighted in orange can help to answer specific material related questions, however, the effort in terms of time, cost or required equipment is often too high for standardised recurring investigations. Grey methods are very elaborate and are mainly meant to address fundamental research questions. Right now, the domain structure is mostly considered in academia (Bitter imaging [64], neutron grating interferometry [66]), however, if reference Bitter imaging patterns of specific characteristics, such as high residual stresses, unfavourable orientations or grain boundaries, can be identified, it could become a great tool to quickly access qualitative data about the final magnetic properties based on the microstructure and texture. The domain structure could potentially link the microstructure and texture to the magnetic properties.

While dealing with NO electrical steel it should always be considered that the name is quite misleading. NO suggests that all properties are similar in all directions, at least within the sheet plane, however, this is not the case. In the final state, NO electrical steel often still has preferred orientations (Figure 3c,d) as well as a differences in texture and grain size along the sheet thickness. These factors result in slightly anisotropic mechanical and magnetic properties which have to be kept in mind.

In Section 3.1, Section 3.2 and Section 3.3, the well-established methods of light microscopy, Vickers hardness testing and X-ray diffraction texture measurements are described, all of which can be used to characterize electrical steel along its process chain (hot rolling, cold rolling, annealing, blanking). As every process step has an influence on the subsequent step and the final microstructure, texture and magnetic properties, these methods should be applied throughout. By including the magnetic property measurements (Section 3.5), we arrive at a set of methods that form both a core characterization ensemble for industrial work and a broad field of research on electrical steel. However, as described above, if applied in a specific manner, the use of these methods must not be restricted to the monitoring of processes or process–property relationships, but can also be employed to address more elaborate research questions, such as how microstructure, texture or alloying elements affect recrystallization or how materials react magnetically to different excitation wave forms, frequencies or higher field strengths. To go beyond these aspects of the characterization of electrical steel and aim at an even deeper understanding of the underlying physical processes, several more advanced methods are now available and outlined in Section 4 of this review. With each of these methods, even though they might not already be established in a given laboratory or require specialist equipment in some cases, one can dive progressively further into the materials physics of electrical steel, particularly if a combination of methods is selected purposefully to answer the research questions at hand. To aid with this selection of methods and to allow a preliminary estimate of the effort and the results associated with a given method, we have aimed to summarise this information for readers who have not used these methods before in Table 4. For example, the use of smaller scale testing (Section 4.1, Section 4.2 and Section 4.3) allows us to relate material behaviour to very specific conditions and meaningful local measurements, such as neutron grating interferometry (Section 4.6) or vector hysteresis measurements (Section 4.4). With these synthesis and characterization methods, we can achieve a comprehensive and interconnected research approach to address fundamental research questions around how deformation structures, grain boundaries, domain structures and magnetic properties are correlated to each other in the controlled framework of single, bi- and oligo-crystals—information that, to the best of the authors’ knowledge, is still largely missing in systematic manner. In addition, there are still open questions about the old—but not fully understood—process of nucleus formation and growth during recrystallization, which may be addressed in a more efficient manner by the methods outlined in Section 4.7 and Section 4.8 and perhaps contribute to a more realistic big picture of the properties of grain boundaries with their high variability in character. Moreover, as discussed in the next section, several of the experimental results can be used to inform, calibrate and optimise process simulation models.

### 5.2. Important Parameters for Simulation Models

Many of the characterization methods described above provide interesting material information for physically based simulation models. This information can either be used for validation or as input parameters. Furthermore, it should be borne in mind that some fundamental research questions must first be answered before fully physically based models become truly possible (nucleus formation, crystal plasticity etc.). In the comment column of Table 4 the bullet point “Simulation” indicates that useful information can be gained for the respective models in the “Relevant Process” column. In the case of light microscopy, the grain morphology development can be tracked along the process chain, some information can be gained about deformation structures and of course the grain size can be measured. All these points can either be input parameters or validation parameters for the respective models along the process chain (rolling, final annealing, blanking, application-magnetic performance). Hardness measurements can be a useful tool to validate the recrystallization kinetics and recrystallization-start temperature for models, taking recovery and recrystallization into account (hot rolling, final annealing). Moreover, it can be used to estimate the input dislocation density (driving force) for recrystallization. For the determination of recrystallization kinetics, in-situ XRD measurements could also be used. Ex-situ texture results can either be used as input parameters or for validation of rolling, annealing and application models, which take the texture development or influence into account. Most deformation models (rolling, blanking) need a macroscopic flow stress evaluation, which can be provided through (stack layer) compression tests at different temperatures and strain rates. However, when the model goes a little deeper and is based on crystal plasticity, the deformation behaviour of single to oligo-crystals (validation) and the critical resolved shear stress of single crystals (input) become important and can be determined through micromechanics. For the validation of application models, the calculated magnetic properties can be compared to measurements from a single-sheet tester in different directions [28]. Two methods which are very important for physically based models concerned with recrystallization (hot rolling, final annealing) are recrystallization scratch experiments and quasi-in-situ EBSD. Thereby, determined grain boundary mobilities can be used directly as input parameters (hot rolling, final annealing) and the remaining results will help us to better understand phenomena like nucleus formation in order to incorporate them later into these models as well. The use of both as direct, physically measured inputs would greatly reduce the amount of parameter fitting and, therefore, uncertainty when comparing with or using for process designs that have in mind a tailor-made material.

## 6. Conclusions

Innovation is often enabled by characterization methods. These can be new methods, improved methods or long-established methods that are used in a new way. In this work, recently developed as well as established characterization methods have been explored with regard to the process chain of NO electrical steel. As the requirements for NO electrical steels are heavily dependent on their respective applications, a good understanding of the interrelationships between processing steps, the phenomena taking place and final material properties is essential to eventually improve the efficiency of an electrical machine. In this context, several different characterization methods, from simple assessments of microstructural state over advanced measurements of physical parameters governing microstructure evolution to new methods for the determination of application properties, have been classified, evaluated and their respective relevance has been underlined with new results. We believe that, together, many of these will serve as a good foundation or even a new starting point to further investigate and understand the interrelationships and phenomena along the process chain of NO electrical steel.

## Figures and Tables

**Figure 1 materials-15-00032-f001:**
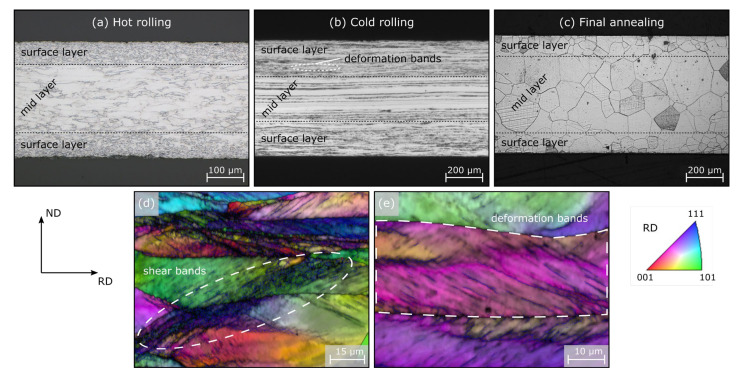
Cross-section (rolling direction (RD)-normal direction (ND) plane) light microscopy images of (**a**) a hot-rolled sheet with some small grains close to the surface (dynamic/static recrystallization), (**b**) a cold-rolled sheet with strongly elongated grains and increased deformation close to the surface (deformation bands) compared to the mid layer, (**c**) a final annealed sheet with small grains close to the surface and larger grains in the mid layer, (**d**,**e**) close-up EBSD IPF || RD maps of heavily deformed areas after cold rolling (d) showing weakly indexed shear bands crossing grain boundaries and (**e**) showing deformation bands stopping at grain boundaries. Note: IPF—inverse pole figure.

**Figure 2 materials-15-00032-f002:**
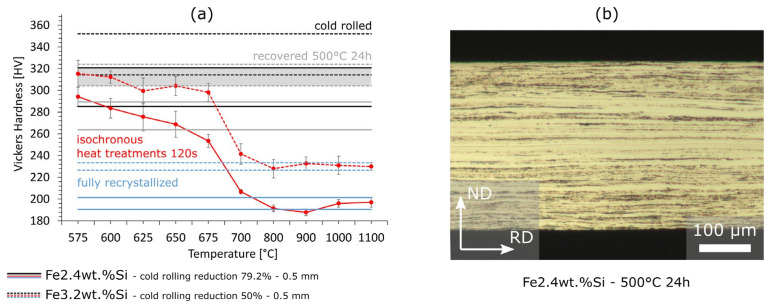
(**a**) Determination of the recrystallization (RX) start temperatures through hardness measurements after isochronal heat treatments for 120 s. The cold-rolled and fully recrystallized and recovered states (24 h at 500 °C) are taken as reference values; dashed lines belong to Fe3.2wt.%Si (cold rolling reduction 50%, final sheet thickness 0.5 mm) (Table 2), solid lines belong to Fe2.4wt.%Si (cold rolling reduction 79.2%, final sheet thickness 0.5 mm), (**b**) cross-sectional view (RD: rolling direction; ND: normal direction) of the 500 °C 24 h recovered Fe2.4wt.%Si sample.

**Figure 3 materials-15-00032-f003:**
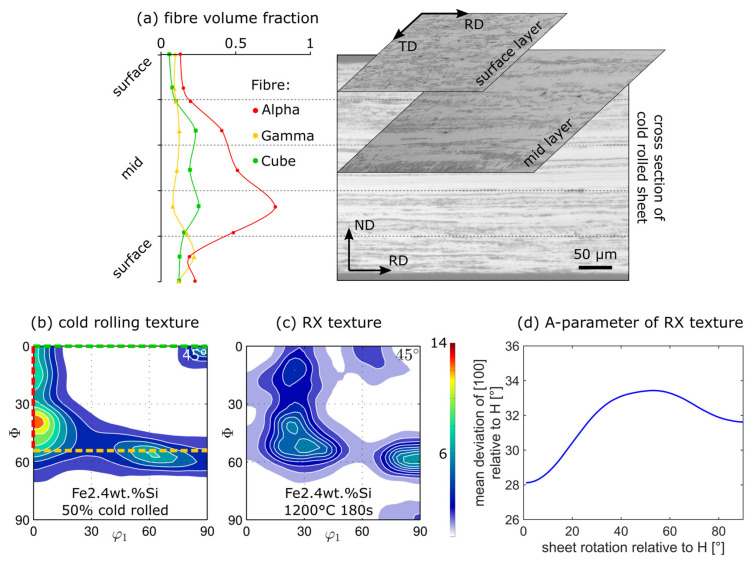
(**a**) Texture fibre distribution along the sheet thickness of a cold-rolled sheet with exemplary light microscopy pictures of the cross-section and in-plane sections in the surface and mid-layer plane (RD/ND/TD: rolling/normal/transverse direction), (**b**) typical cold rolling (Fe2.4wt.%Si 50% cold rolled to 0.5 mm) and (**c**) recrystallization (RX) texture (material from (**b**) after final annealing: 1200 °C 180 s) composed of at least a surface- and a mid-layer measurement shown in a φ_2_ = 45° section of an ODF and (**d**) A-parameter of (**c**).

**Figure 4 materials-15-00032-f004:**
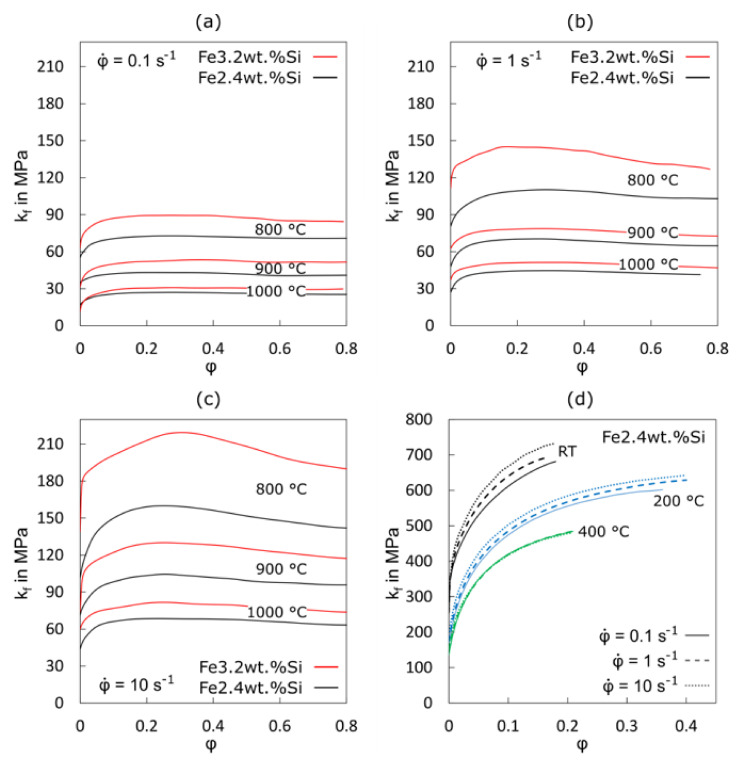
(**a**–**c**) Representative hot rolling flow curves determined with the help of cylindrical compression tests at different strain rates and temperatures for Fe2.4wt.%Si and Fe3.2wt.%Si and (**d**) average cold-to-warm rolling flow curves (based on three tests each) determined with the help of cylindrical stack layer compression tests at different strain rates and temperatures for Fe2.4wt.%Si.

**Figure 5 materials-15-00032-f005:**
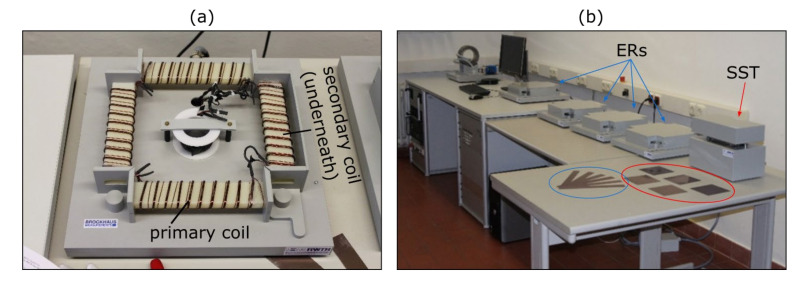
(**a**) Epstein frame (ER) with primary and secondary coils around the four limbs, (**b**) lab with four ERs and one single-sheet tester (SST) as well as several ER strip samples (blue circle) and several rectangular SST samples (red circle).

**Figure 6 materials-15-00032-f006:**
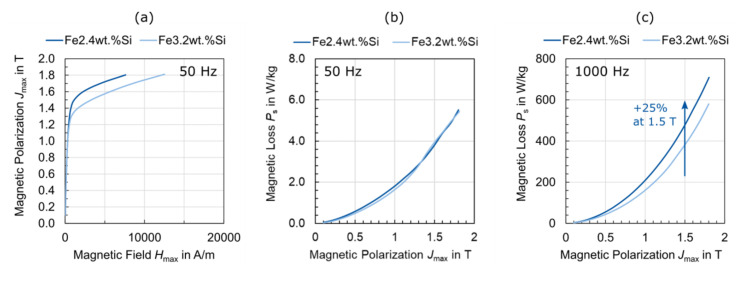
Magnetic properties measured with a single-sheet tester (SST) of two electrical steel grades with different silicon contents (Fe2.4wt.%Si and Fe3.2wt.%Si), but similar sheet thickness (0.5 mm), texture and grain size (224 µm and 231 µm), (**a**) magnetic polarization at 50 Hz, (**b**) magnetic loss at 50 Hz and (**c**) magnetic loss at 1000 Hz.

**Figure 7 materials-15-00032-f007:**
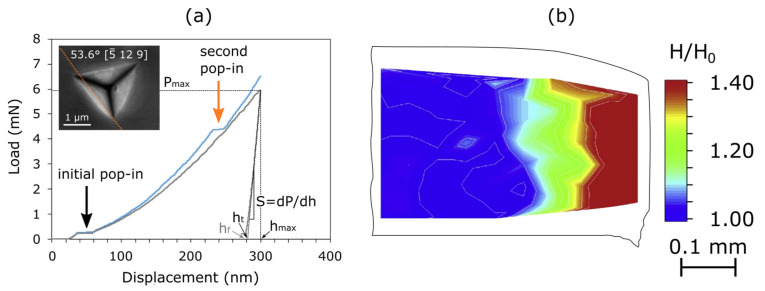
(**a**) Exemplary load displacement curves of two nanoindents in a Fe2.4wt.%Si sheet close to a mixed 53.6° [
$\bar{5}\;12\;9]$
 grain boundary with predominantly low transmission factors, one curve showing only a first pop-in (grey—with indicated values (*h*-depth, *P*-load) for hardness or Young’s modulus *S* calculations, see [41] for more information) and one curve showing a first and a second pop-in (blue), (**b**—reprinted with permission from the authors of [47], © 2021 Elsevier) cross-section close to the cutting edge of a Fe2.4wt.%Si sheet with a microindentation array (a nanoindentation array would increase the resolution heavily) coloured according to the respective relative hardness values.

**Figure 8 materials-15-00032-f008:**
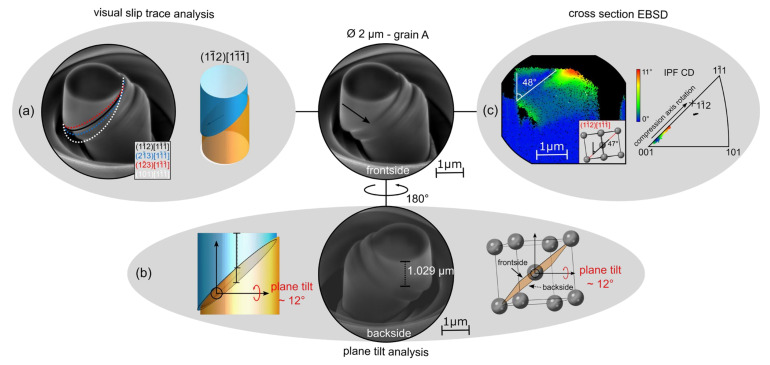
Exemplary slip trace analysis approaches, (**a**) visual slip trace analysis, (**b**) plane tilt analysis and (**c**) cross-section EBSD. Reprinted with permission from the authors of [51], © 2021 Creative Commons CC BY 4.0.

**Figure 9 materials-15-00032-f009:**
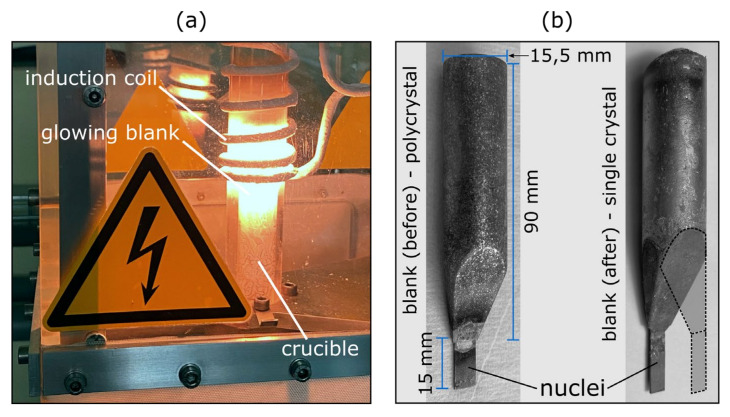
(**a**) Close-up picture of the self-built induction furnace with a built-in crucible and glowing blank, (**b**) blank before (polycrystal) and after (single crystal) crystal growth, whereby, in the latter picture, the bi-crystal geometry is also depicted (dashed lines).

**Figure 10 materials-15-00032-f010:**
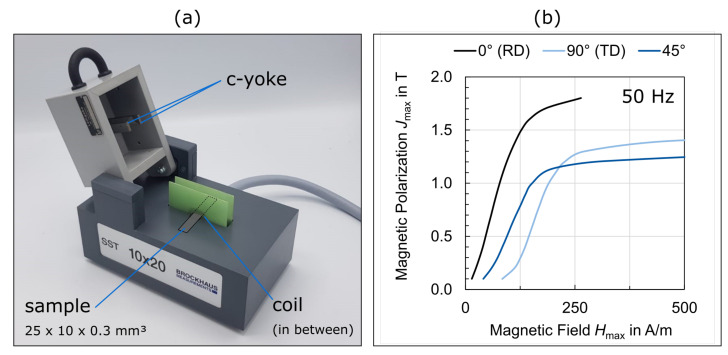
(**a**) Miniature single-sheet tester with a schematic sample and (**b**) corresponding polarization curves at 50 Hz of a grain-oriented electrical steel magnetized in different directions relative to the initial rolling direction (RD).

**Figure 11 materials-15-00032-f011:**
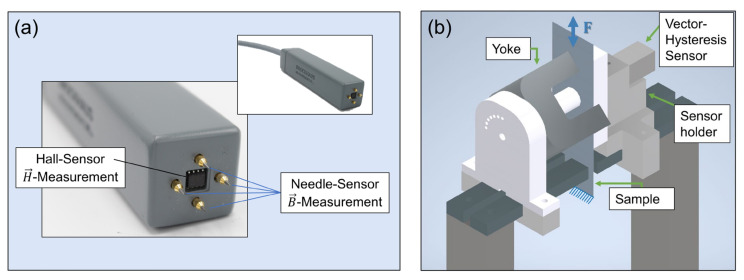
(**a**) Vector hysteresis sensor and (**b**) schematic setup to investigate vector resolved magnetic properties with a vector hysteresis sensor while applying a tensile stress (DFG SPP 2013 unpublished work [62]).

**Figure 12 materials-15-00032-f012:**
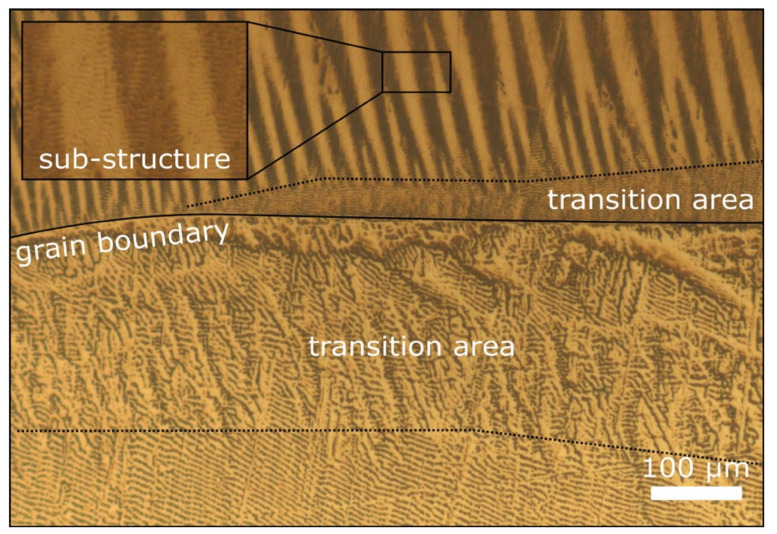
Bitter image of a slightly deformed Fe2.4wt.%Si bi-crystal with a pattern transition close to the grain boundary using a commercial water-based ferrofluid with a particle diameter of 10 nm (EMG 508, FerroTec: Santa Clara, CA, USA).

**Figure 13 materials-15-00032-f013:**
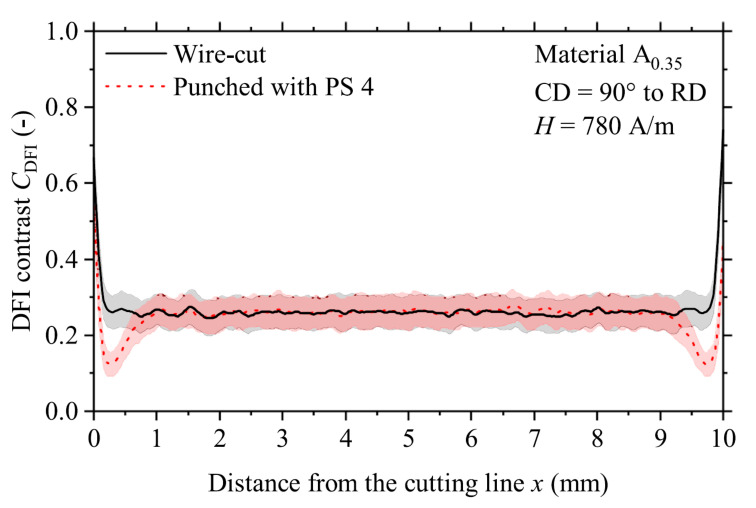
DFI contrast (C_DFI_) of a wire-cut and punched sample. Reprinted with permission from the authors of [76], © 2021 Elsevier.

**Figure 14 materials-15-00032-f014:**
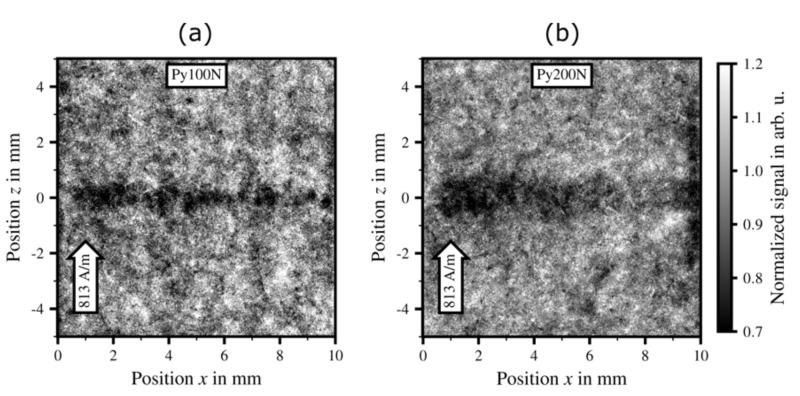
Exemplary neutron grating interferometry results of two Fe2.4wt.%Si samples being magnetized at 813 A/m and embossed along a line (position z = 0) with a pyramidal punch with a force of (**a**) 100 N and (**b**) 200 N. Reprinted with permission from the authors of [75], © 2021 Creative Commons CC BY.

**Figure 15 materials-15-00032-f015:**
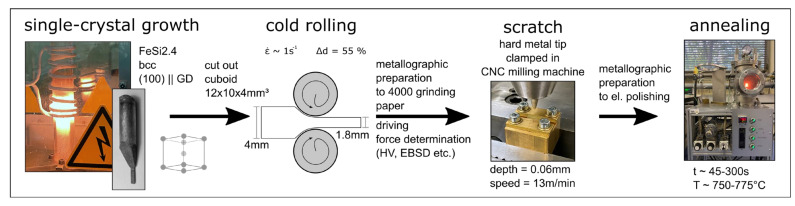
Schematic experiment layout for the determination of grain boundary mobilities on a statistical basis.

**Figure 16 materials-15-00032-f016:**
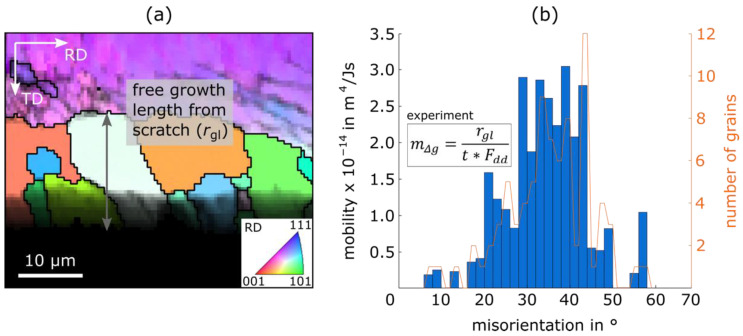
(**a**) Exemplary section from an EBSD panorama image heat-treated at 765 °C for *t* = 300 s, with one free growth length *r*_gl_ from the scratch drawn in, coloured according to an IPF || RD and (**b**) results of experimental grain boundary mobility calculations (plus equation) based on over 80 grains divided into misorientation classes Δ*g*, with a width of 2°. IPF—inverse pole figure.

**Figure 17 materials-15-00032-f017:**
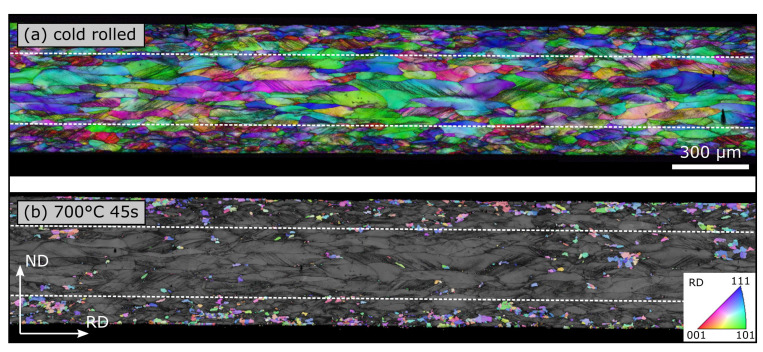
Quasi-in-situ edge (normal direction (ND), rolling direction (RD)) EBSD panorama excerpts of the same area, (**a**) before and (**b**) after annealing (700 °C 45 s), whereby the microstructure in (**a**) and isolated nuclei (<1.5 ° GOS) in (**b**) are coloured according to an IPF || RD, while the background is based on image quality values. Note: IPF—inverse pole figure.

**Table 1 materials-15-00032-t001:** Correlations between different parameters and magnetic as well as mechanical properties; (red) poorer performance; (green) superior performance; (grey) no direct influence; (circle section) higher (red) or lower (green) relative share of total iron loss.

Parameter		Classical Loss (P_cl_)	Hysteresis Loss (P_h_)	Excess Loss (P_ex_)	Magnetizability (J)	Yield Strength (σ_y_)
Frequency (total iron loss increases)	↑	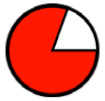	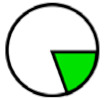	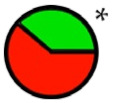		Increased requirements
Electrical resistivity (% Si, Al, Mn)	↑	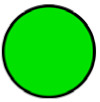			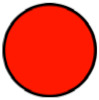	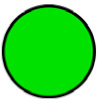
Grain size	↑		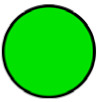	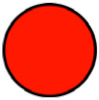	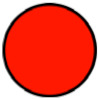	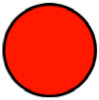
Beneficial texture (low A-parameter)			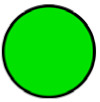		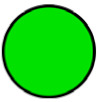	
Sheet thickness	↓	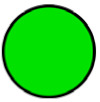	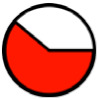	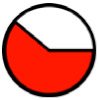		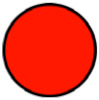
Residual stresses	↓		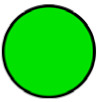	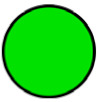	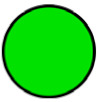	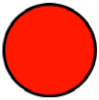

* increasing share with lower and decreasing share with higher sheet thickness, performance: poor
superior.

**Table 2 materials-15-00032-t002:** Chemical analysis: Fe2.4wt.%Si—provided by supplier, Fe3.2wt.%Si—measured by spark spectral analysis (S, N, P and C are either overestimated or below the detection limit of the method, therefore, the maximum content based on supplier information (*) is provided).

Material	Share in wt.%							
	Si	Al	Mn	S	N	P	C	Fe
Fe2.4wt.%Si	2.38	0.36	0.17	0.001	0.002	0.012	0.002	balance
Fe3.2wt.%Si	3.16	0.89	0.17	* 0.003	* 0.006	* 0.07	* 0.002	balance

**Table 3 materials-15-00032-t003:** Material specific parameters associated with the flow curve model by Hensel and Spittel [37] for Fe2.4wt.%Si and Fe3.2wt.%Si.

Material	A	m_1_	m_2_	m_3_	m_4_	m_5_	m_7_	m_8_	m_9_
Fe2.4wt.%Si	2.08∙10^10^	17.8∙10^−4^	0.092	0.050	2.050∙10^−5^	2∙10^−4^	−0.376	1.4∙10^−4^	−2.613
Fe3.2wt.%Si	3.35∙10^17^	9.6∙10^−4^	0.071	0.094	5.907∙10^−6^	4∙10^−4^	−0.469	9.9∙10^−5^	−5.397

**Table 4 materials-15-00032-t004:** (green) Great input/output ratio—always recommended; (orange) medium input/output ratio—recommended for certain questions; (grey) fundamental research, the more plusses the more elaborate the method, “Preparation x µm” indicates the minimum final metallographic preparation step for the respective method, “Simulation” indicates that the respective results can be useful for process modelling.

Analysis Method	Relevant Process	Effort	Insight	Comment
light microscopy (Section 3.1)	hot rolling cold rolling final annealing blanking application	++	grain shape deformation structures (mesoscopic) grain size (ASTM E112-13)	layer specific cross-section for overview preparation 1 µm etchant 5% Nital simulation
Vickers hardness (Section 3.2)	hot rolling cold rolling final annealing blanking application	+	driving force recrystallization progress work hardening Hall–Petch segregation	preparation 1 µm settings 15 s 0.2 N spacing 10× depth Simulation DIN 6507-1
X-ray diffraction (Section 3.3)	hot rolling cold rolling final annealing application	++	texture A-parameter recrystallization progress or kinetics deformation (peak width)	preparation 1 µm area 8 mm × 8 mm RD important layer-specific simulation
tensile test	cold rolling blanking application	+	flow stress mechanical properties	specific geometry DIN 6892-1
stack layer compression test (Section 3.4)	cold rolling blanking	++	flow stress	specific geometry Simulation
compression test (Section 3.4)	hot rolling	+	flow stress	specific geometry ASTM E9-19
single-sheet tester (Section 3.5)	final annealing blanking application	+	magnetic properties	specific geometry unidirectional simulation DIN EN 60404
Epstein frame (Section 3.5)	final annealing blanking application	++	magnetic properties	specific geometry unidirectional DIN EN 60404
micromechanics (nanoindentation, micropillar compression) (Section 4.1)	cold rolling blanking application	+++	deformation behaviour	preparation OPU FIB milling nanoindenter (with flat punch) simulation
crystal growth (Section 4.2)	blanking application	+++	crystal lattice growth behaviour	special geometry preliminary work for: miniature single-sheet tester, Bitter imaging, recrystallization scratch experiment
recrystallization scratch experiment (Section 4.7)	final annealing	+++	grain boundary mobility	single crystal preparation el. polishing simulation
quasi-in-situ EBSD (Section 4.8)	final annealing	+++	deformation structure nucleus formation grain growth	cross-section preparation el. polishing simulation
neutron grating interferometry (Section 4.6)	blanking application	+++++	local domain structure deformation structures	special geometry rarely available
miniature single-sheet tester (Section 4.3)	blanking application	+	magnetic properties	crystal growth needed special geometry unidirectional unique machine
Bitter imaging (Section 4.5)	blanking application	++	domain structure	not yet fully explored EMG 508
vector hysteresis sensor (Section 4.4)	final annealing blanking application	+	local magnetic properties	facilitates the measurement of multidirectional properties

## Data Availability

The datasets generated and analyzed during the current study are available from the corresponding author on reasonable request.

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
