# Peer review of "Characterization Methods along the Process Chain of Electrical Steel Sheet—From Best Practices to Advanced Characterization"

_materials, 2021, doi:10.3390/ma15010032_

Round 1

Reviewer 1 Report

This paper introduces non-oriented (NO) electrical steel sheets and their applications to various engineering structures. Several experimental results are reviewed to show the strengths of the different methods as well as their (dis)advantages, typical applications and obtainable data. Basically, the paper is well prepared and has a good structure. Hence the manuscript is recommended for publication with minor revisions.

1) It would be better to cite the related references after several typical design strategies, since these strategies are already proposed in some published papers.

2) It would be useful to give a content at the beginning of this paper.

3) It is necessary to explain the meaning of the grey regions in Table 1.

Author Response

Dear Reviewer,

Thank you very much for your time and valuable comments. Please find our answers below in bold font.

This paper introduces non-oriented (NO) electrical steel sheets and their applications to various engineering structures. Several experimental results are reviewed to show the strengths of the different methods as well as their (dis)advantages, typical applications and obtainable data. Basically, the paper is well prepared and has a good structure. Hence the manuscript is recommended for publication with minor revisions.

1) It would be better to cite the related references after several typical design strategies, since these strategies are already proposed in some published papers.

Answer: We agree and have moved several original papers closer to the first mention of the respective methods (see below). From this, we excluded light microscopy, Vickers hardness, XRD, SST, ER and EBSD, as they are well established and partly already laid down in norms, as well as the miniature SST as this is a new method.

Added references:

  • Line 381 - Flow curves in general
    • Line 393 - isothermal hot compression tests
    • Line 413 - stack layer compression tests
  • Line 543 - nanoindentation
  • Line 543 - micropillar compression
  • Line 658 - Crystal growth
  • Line 738 - Vector hysteresis sensor
  • Line 758 - Bitter imaging
  • Line 802 - Neutron
  • Line 895 - Scratch
  • Line 1006 - ECCI

2) It would be useful to give a content at the beginning of this paper.

Answer: Thank you for this suggestion. Unfortunately, we were not entirely sure whether the reviewer referred to a table of contents or the content of alloying elements. Either way, both suggestions are very good, so we have included both, a table of content and the chemical composition at the end of section 1. (Lines 117-119)

3) It is necessary to explain the meaning of the grey regions in Table 1.

Answer: Thank you, that is a good point, we have changed the caption accordingly and included “(grey) no direct influence”. (Line 86)

Reviewer 2 Report

“Characterization methods along the process chain of electrical steel sheet - from best practices for standard methods to advanced characterization”.

Generally speaking, a manuscript of a review nature. Due to the fact that in many cases the information provided by the authors is only the basis of research techniques generally used by researchers, the assessment of the content of the manuscript may raise some doubts. This state of affairs mainly concerns hardness measurements (including micro- and nano-), metallographic tests using light and electron microscopy, and X-ray diffraction tests. Nevertheless, I believe that the list and characteristics of selected research methods proposed by the authors in the form of a comprehensive study constitute an interesting and valuable approach to the issues of structural research on materials. Below are some comments that are worth paying attention to.

  1. In my opinion, the research techniques presented in the manuscript are not sufficiently interconnected. When reading the content of the manuscript, one gets the impression that its character is only textbook, i.e. without drawing conclusions regarding the assessment of the tested material, the properties of which were designed in the course of the exemplary thermo-mechanical treatment process. I understand that the nature of the manuscript has been oriented towards presenting research techniques, however, in this form it is not a clearly defined research paper from which other researchers could derive specific indicators describing the behavior of the steels presented in the manuscript. Therefore, it is suggested that in the chapters on individual research methods, more attention should be paid to the evaluation of the obtained research results, than to the theoretical aspects of the research techniques themselves. For example, in chapter 3.1 the authors presented methods of revealing the microstructure of the material after the rolling process, however, the obtained results were not assessed, especially since a specific steel grade was tested.
  2. In the final part of the manuscript, there was no qualitative discussion of the results obtained. In their work, the authors presented two grades of steel that were subjected to specific technological operations with the use of various research and measurement equipment. However, no conclusions regarding the obtained results were presented. Therefore, it is suggested that the authors conduct a discussion - regardless of the research issue itself - regarding the behavior of the presented materials, the properties of which have been "shaped" in various technological operations.
  3. Lines 278, 443, 706, 920: Are the geometrical dimensions of the samples specified correctly? In the case of units, I guess it should be "mm", no any superscript.

Overall, the work is very interesting and valuable. The few comments that can be made to the content of the manuscript are that the results of the cited experiments are not related to each other. In its present form, the work gives the impression of only a literature description of various research methods that can actually be used independently, without creating a comprehensive research process. Thus, such a discussion, e.g. in a summary, would be appreciated.

Author Response

Dear Reviewer,

Thank you very much for your time and valuable comments. Please find our answers below in bold font.

“Characterization methods along the process chain of electrical steel sheet - from best practices for standard methods to advanced characterization”.

Generally speaking, a manuscript of a review nature. Due to the fact that in many cases the information provided by the authors is only the basis of research techniques generally used by researchers, the assessment of the content of the manuscript may raise some doubts. This state of affairs mainly concerns hardness measurements (including micro- and nano-), metallographic tests using light and electron microscopy, and X-ray diffraction tests. Nevertheless, I believe that the list and characteristics of selected research methods proposed by the authors in the form of a comprehensive study constitute an interesting and valuable approach to the issues of structural research on materials. Below are some comments that are worth paying attention to.

  1. In my opinion, the research techniques presented in the manuscript are not sufficiently interconnected. When reading the content of the manuscript, one gets the impression that its character is only textbook, i.e. without drawing conclusions regarding the assessment of the tested material, the properties of which were designed in the course of the exemplary thermo-mechanical treatment process. I understand that the nature of the manuscript has been oriented towards presenting research techniques, however, in this form it is not a clearly defined research paper from which other researchers could derive specific indicators describing the behavior of the steels presented in the manuscript. Therefore, it is suggested that in the chapters on individual research methods, more attention should be paid to the evaluation of the obtained research results, than to the theoretical aspects of the research techniques themselves. For example, in chapter 3.1 the authors presented methods of revealing the microstructure of the material after the rolling process, however, the obtained results were not assessed, especially since a specific steel grade was tested.

Answer: Thank you for this comment. As the reviewer correctly pointed out, the focus of this article is and was intended to be on the characterization methods themselves and how these can contribute to open research questions, simulation models and process control. The reason for this is that this article is part of the special issue “Low-Loss Non-Oriented Electrical Steel Sheet for Energy-Efficient Electrical Drives” together with four additional articles. Thereby, every article focuses on a different aspect of the exact same material. For example DOI: 10.3390/ma14226822 resolves around the influence of process parameters on grain size and texture evolution during hot/cold rolling and annealing. Considering the complete special issue, we believe that the materials have been characterized, described and discussed in great detail. With a too detailed description/discussion of the characterization results in the present work, we would run the risk of repeating ourselves and the paper would become even more extensive. Therefore, we only touch upon measurement results and material behavior and try to keep most of the data generally valid for electrical steel, while still providing new data that we believe is valuable for the community (especially section 3.4 - material dependent parametrized flow stress model, section 3.5 - material & frequency dependent magnetic properties, section 4.3 - orientation dependent magnetic properties, section 4.7 – material dependent grain boundary mobility data).

However, we do realize that, just like the reviewer now, potential readers may very well not be aware of this connection of the special issue, even after publication. We have therefore now included an extra paragraph explicitly pointing to the connected publications that address the point made by the reviewer. We hope that this is an acceptable compromise, particularly as all these manuscripts have now already been accepted and are all available open access to interested readers.

The new paragraph has been inserted in lines 93-100

“The work at hand is part of the special issue “Low-loss non-oriented electrical steel sheet for energy-efficient electrical drives”, in whose publications the same material was used throughout (see chemical composition in Table 2). Thereby, every publication focuses on a different aspect of this material. Two papers revolve around the processing and its influence on the microstructure, texture and magnetic properties [4,5], one deals with integrated process simulation [6] and the last one with material design [3], all drawing heavily on experimental results from the established and advanced methods described and discussed here.”

  1. In the final part of the manuscript, there was no qualitative discussion of the results obtained. In their work, the authors presented two grades of steel that were subjected to specific technological operations with the use of various research and measurement equipment. However, no conclusions regarding the obtained results were presented. Therefore, it is suggested that the authors conduct a discussion - regardless of the research issue itself - regarding the behavior of the presented materials, the properties of which have been "shaped" in various technological operations.

Answer: We would like to refer to the previous answer here, this type of discussion is indeed included in the other publications and we would like to maintain a focus here on the review character of the manuscript and not mix this information with a discussion referring to a special alloy too much beyond the examples given in the text and figures.

  1. Lines 278, 443, 706, 920: Are the geometrical dimensions of the samples specified correctly? In the case of units, I guess it should be "mm", no any superscript.

Answer: Thank you for reading the manuscript so carefully, we have changed all the units to avoid misunderstandings.

Overall, the work is very interesting and valuable. The few comments that can be made to the content of the manuscript are that the results of the cited experiments are not related to each other. In its present form, the work gives the impression of only a literature description of various research methods that can actually be used independently, without creating a comprehensive research process. Thus, such a discussion, e.g. in a summary, would be appreciated.

Answer: We agree that the interconnection between the sections can be improved and added the following paragraph to the discussion (Lines 1060-1095):

“In section 3.1 to 3.3, the well-established methods of light microscopy, Vickers hardness testing and X-ray diffraction texture measurements are described, all of which can be used to characterize electrical steel along its process chain (hot rolling, cold rolling, annealing, blanking). As every process step has an influence on the subsequent step and the final microstructure, texture and magnetic properties, these methods should be applied throughout. By including the magnetic property measurements (section 3.5), we arrive at a set of methods that form a core characterization ensemble for industrial work and a broad field of research on electrical steel alike. However, as described above, if applied in a specific manner, the use of these methods must not be restricted to monitoring of processes or process-property-relationships, but can also be employed to address more elaborate research questions, such as how microstructure, texture or alloying elements affect recrystallization or how materials react magnetically to different excitation wave forms, frequencies or higher field strengths. To go beyond these aspects of the characterization of electrical steel and aim at an even deeper understanding of the underlying physical processes, several more advanced methods are now available and outlined in section 4 of this review. With each of these methods, even though they might not already be established in a given laboratory or require specialist equipment in some cases, one can dive progressively further into the materials physics of electrical steel, particularly if a combination of methods is selected purposefully to answer the research questions at hand. To aid with this selection of methods and to allow a preliminary estimate of the effort as well as the results associated with a given method, we have aimed to summarise this information for readers who have not used these methods before in Table 4. For example, the use of smaller scale testing (sections 4.1 to 4.3) allows us to relate material behaviour to very specific conditions and meaningful local measurements, such as neutron grating interferometry (section 4.6) or vector hysteresis measurements (section 4.4). With these synthesis and characterization methods, we can achieve a comprehensive and interconnected research approach to address fundamental research questions around how deformation structures, grain boundaries, domain structures and magnetic properties are correlated to each other in the controlled framework of single, bi- and oligo-crystals. Information that, to the best of the authors' knowledge, is still largely missing in systematic manner. In addition, there are still open questions about the old but not fully understood process of nucleus formation and growth during recrystallization, which may be addressed in a more efficient manner by the methods outlined in sections 4.7 and 4.8 and perhaps contribute to a more realistic big picture of the properties of grain boundaries with their high variability in character. Moreover, as discussed in the next section, several of the experimental results can be used to inform, calibrate and optimise process simulation models.”

As mentioned above, this work needs to be seen in context of the special issue. The relationship between the experiments in this paper is that all methods are highly relevant for electrical steel, whereby the results can be used for characterization, simulation or fundamental research. The methods themselves can be used independently for respective research questions. The paper should be a toolbox for further research, while still presenting new and in our opinion valuable data in many sections. Moreover, we want to highlight that there are still a lot of open topics in electrical steel although the tools to access them are already available.

Reviewer 3 Report

Regardless of the effort and good intentions of the author, unfortunately I state some facts as follows:
1- Title of the paper is too long
2. Most part of the abstract belongs to the introduction.
3. English language and sentence constructions below the allowed level.
4. I find nothing new or interesting rash repetition of familiar details
5. This is work that belongs to the lower level of a congress paper
6. The work is in any case outside the domain of the reputation of the first quartile journal in terms of its quality and manner of writing.
I do not recommend work for further consideration

Author Response

Dear Reviewer,

Thank you for your time and valuable comments. Please find our answers below in bold font.

Regardless of the effort and good intentions of the author, unfortunately I state some facts as follows:

  1. Title of the paper is too long

Answer: Thank you for this remark, we tried to shorten the title without losing its expressiveness:

“Characterization methods along the process chain of electrical steel sheet - from best practices to advanced characterization”

  1. Most part of the abstract belongs to the introduction.

Answer: You are right, the abstract is rather general. However, this is due to the structure and idea of the paper. We would like to highlight open research questions in the field of electrical steel and how to approach them with different methods while still providing interesting results for most of the presented methods. As there are many individual experimental highlights we cannot mention all of them in the abstract. Therefore, we tried to introduce the topic, underline its relevance and give a distinct description of what to expect from the paper.

  1. English language and sentence constructions below the allowed level.

Answer: The manuscript has been proofread by several researchers without major deficiencies being found. Nevertheless, minor corrections were made. If the reviewer could mention some examples we would be happy to further improve the manuscript.

  1. I find nothing new or interesting rash repetition of familiar details

Answer: We aim to provide a toolbox/starting point for researchers joining the field of electrical steel (Table 4), potentially saving time and cost. Furthermore, we believe that especially section 4 (Advanced methods) is beyond the common knowledge of material scientists. We have tried to highlight that there are still many open questions in this field and would like to stimulate further research. Moreover, we present new and in our opinion interesting data regarding Fe2.4wt.-%Si & Fe3.2wt.-%, especially in the following sections:

  • Section 3.2 Figure 2. - results/discussion of material & processing dependent recrystallization kinetics
  • Section 3.3 Figure 3. - total and layer dependent texture distribution as well as A-parameter
  • Section 3.4 Figure 4. & Table 3 - material dependent flow stress data
  • Section 3.5 Figure 6 - frequency and material dependent magnetic properties
  • Section 4.3 Figure 10 - texture related magnetic properties of oligo-crystals
  • Section 4.7 Figure 16 - material dependent grain boundary mobility data
  1. This is work that belongs to the lower level of a congress paper

Answer: We are not sure how to approach this comment. We conducted all experiments according to the guidelines for good academic practice and tried to write a self-contained article that is valuable to the community. We would like to point out that this article is part of the special issue “Low-Loss Non-Oriented Electrical Steel Sheet for Energy-Efficient Electrical Drives” together with four additional articles. Thereby, every article focuses on a different aspect of the exact same material. For example DOI: 10.3390/ma14226822 resolves around the influence of process parameters on grain size and texture evolution during hot/cold rolling and annealing. The present paper should be a toolbox for further research, while still presenting new and in our opinion valuable data in many sections. We tried to highlight flaws of common methods, introduce new methods and use common methods in a new way.

  1. The work is in any case outside the domain of the reputation of the first quartile journal in terms of its quality and manner of writing.

Answer: We have made several changes to the manuscript and hope that the revised version will meet the requirements of the reviewer and journal.

I do not recommend work for further consideration

Round 2

Reviewer 3 Report

Since efforts have been made to repair certain parts of the paper and since the answers of the authors are presented in a very accessible tone, I suggest accepting the paper.